# TLXML: Task-Level Explanation of Meta-Learning via Influence Functions

## Abstract

The scheme of adaptation via meta-learning is seen as an ingredient for solving the problem of data shortage or distribution shift in real-world applications, but it also brings the new risk of inappropriate updates of the model in the user environment, which increases the demand for explainability. Among the various types of XAI methods, establishing a method of explanation based on past experience in meta-learning requires special consideration due to its bi-level structure of training, which has been left unexplored. In this work, we propose influence functions for explaining meta-learning that measure the sensitivities of training tasks to adaptation and inference. We also argue that the approximation of the Hessian using the Gauss-Newton matrix resolves computational barriers peculiar to meta-learning. We demonstrate the adequacy of the method through experiments on task distinction and task distribution distinction using image classification tasks with MAML and Prototypical Network.

## 1 Introduction

Meta-learning is a widely studied class of methods that enables models to quickly adapt to unseen tasks (Finn et al., 2017), addressing limitations in generalization due to data scarcity during training phase (Song & Jeong, 2024; Li et al., 2018; Shu et al., 2021; Lu et al., 2021) or distribution shift in the user environment (Mann et al., 2021; Mouli et al., 2024; Lin et al., 2020). Although meta-learning methods guarantee faster adaptation than conventional methods, they do not account for *safe* adaptations (Zhang et al., 2020). In fact, they may cause inappropriate model updates in user environments, which raises safety concerns in downstream tasks (Khattar et al., 2024; Wen et al., 2022; Yao et al., 2024). Consequently, This increases the need for explanation methods to enhance transparency and ensure the safe operation of autonomous systems.

The most commonly proposed explanation methods in this decade focus on local explanation: They treat the model parameters as given and explain the model's local behavior around specific inference data points. Although these methods can be applied to the models with their weights determined via meta-learning and adaptation, they do not cover the needs raised by the peculiarities of meta-learning (Figure 1a). Because they are based solely on the input data used at each inference, what they can achieve at most is assessing and comparing the importance of the components in a single data, and understanding those explanations requires knowledge about those components, which is sometimes technical and of expert levels (Adebayo et al., 2022). This requirement is considered to limit the application area where local explanations can resolve the above-mentioned safety concerns of meta-learning.

One way without relying on the details of the inference data is to avoid seeing the lower level structure and instead analyze the relation to previously known data, i.e. training data. In meta-learning, this expectation is supported by the fact that in some cases, where analyses based on training tasks are essential for understanding the model's behaviors (Goldblum et al., 2020b). This insight leads us to consider highlighting how previously encountered tasks have influenced the model's current inference (Figure 1b). Notice that explanations based on past experiences have an advantage in utilizing structures of the training dataset, which increases the conciseness of explanations. In the case of Figure 3b, the second half of the explanation is possible owing to the nature that training data can be supplemented with their attributes with various levels of abstraction, which can be added by annotators, or taken from descriptions of the data acquisition process. Such conciseness enables

users to draw analogies to familiar situations, enhancing their understanding of the new task. This is essential as future real-world applications will likely involve training datasets from diverse domains, making it difficult for non-expert users to understand explanations without abstracting the raw data.

Explanation based on past experiences has been studied for supervised learning (Koh & Liang, 2017; Jeyakumar et al., 2020; Yeh et al., 2019; Wolf et al., 2024). Particularly, Koh & Liang (2017) proposed a method to evaluate the influence of training data points on the model's predictions using influence functions (Hampel, 1974). To achieve this, they introduced a perturbation parameter that upweights the loss contribution of each training data. By calculating the derivative of the model's learned parameters with respect to this parameter, they were able to trace the influence of training data on the model's behavior. Influence functions are effective techniques (Zhang et al., 2022; Koh et al., 2019; Han & Tsvetkov, 2020) widely utilized in standard supervised learning settings, offering powerful tools for understanding model robustness (Cohen et al., 2020), measure the effectiveness of data-models (Saunshi et al., 2022), and improving model interpretability (Chhabra et al., 2024). Nevertheless, utilizing influence functions in meta-learning settings has not been explored.

In this paper, we introduce Task-Level eXplanation of Meta-Learning (TLXML), a novel method that leverages influence functions to quantify the impact of previously learned tasks (training tasks) in a meta-learning framework. The primary objective of this work is to extend the focus of Koh and Liang's approach (Koh & Liang, 2017) to meta-learning. We empirically demonstrate that TLXML quantifies the influence of the training tasks on the meta-parameters, adapted network weights, and inference, which can serve as a similarity measure between training and test tasks.

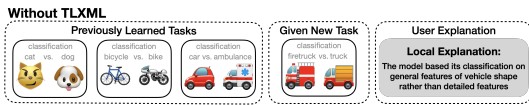

(a) Without TLXML, it is common to explain the model's behavior via local explanations.

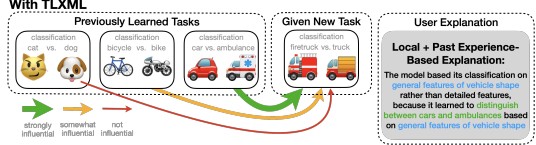

(b) TLXML calculates the influence of each previously learned task on the model's behavior in a given new task, resulting in a more effective user explanation.

Figure 1: Key insights of TLXML.

**Contributions.** We summarize our contributions as follows: 1) Task-Level Explanations for Meta-Learning: We introduce TLXML for assessing the influence of meta-training tasks on adaptation and inference during meta-tests. it provides concise, task-based explanations that align with users' abstraction levels, facilitating better interpretability of meta-learning processes. 2) Computation optimization: we reveal that TLXML, for a network with $p$ weights and $q$ meta-parameters, incurs an expensive computational cost of $O(pq^2)$ in its exact form, which does not scale up to complex networks. We then introduce an approximation method for the Hessian matrix of the training loss using Gauss-Newton matrix, which reduces the cost to $O(pq)$.

## 2 RELATED WORK

**Influence functions for machine learning**. The primary focus of existing research is the use of influence functions in standard supervised learning models initiated by Koh & Liang (2017). Influence functions have been successfully used for multiple purposes, such as explaining model behavior with respect to training data in various tasks (Barshan et al., 2020; Koh & Liang, 2017; Han et al., 2020), quantifying model uncertainty (Alaa & Van Der Schaar, 2020), crafting/detecting adversarial training examples (Cohen et al., 2020).These approaches focus on data-level explanations, rendering them of limited practical value in meta-learning settings. TLXML leverages influence functions for task-level explanations, offering more effective insights into how training tasks influence the model's behavior. Nevertheless, as noted in Alaa & Van Der Schaar (2020), computing the Hessian matrix is expensive. To scale the methods of influence functions to more complex networks and larger datasets. TLXML uses an approximation method for faster computation of the matrix.

**Explainable AI (XAI) for meta-learning**. It is natural to consider applying existing XAI methods to meta-learning. The methods that are agnostic to the learning process should be applicable for explaining inference in meta-learning. Although this area is still in its early stages, some research examples already exist (Woźnica & Biecek, 2021; Sijben et al., 2024; Shao et al., 2023). The closest

work to ours is by Woźnica & Biecek (2021) who quantified the importance of meta-features, i.e., high-level characteristics of a dataset such as size, number of features and number of classes. In meta-learning, the goal is to train models that can generalize across various tasks. We argue that understanding the influence of *training tasks* play an important role in evaluating the model's adaptability, whereas meta-features offer limited insight for such evaluation.

**Impact of training data**. The robustness of machine learning models is another area where the impact of training data on inference is frequently discussed (Khanna et al., 2019; Ribeiro et al., 2016). This topic has also been explored in the context of meta-learning such as creating training-time adversarial attacks via meta-learning (Zügner & Günnemann, 2019; Xu et al., 2021), training robust meta-learning models by exposing models to adversarial attacks during the query step of meta-learning (Goldblum et al., 2020a), and data augmentation for enhancing the performance of meta-learning algorithms (Ni et al., 2021b). However, influence functions have not been utilized for meta-learning in this field. Similar to Khanna et al. (2019); Barshan et al. (2020) in the case of supervised learning, TLXML also leverages Fisher information metrics. This formulation provides a valuable geometric viewpoint for analyzing the model's parameter space.

## 3 PRELIMINARIES

**Influence functions**. Koh & Liang (2017) proposed influence functions for measuring the impact of training data on the outcomes of supervised learning models, under the assumption that the trained weights $\hat{\theta}$ minimize the empirical risk:

$$\hat{\theta} = \underset{\theta}{\operatorname{argmin}} \, L\left(D^{\text{train}}, f_\theta\right) = \underset{\theta}{\operatorname{argmin}} \, \frac{1}{n} \sum_{i=1}^{n} l\left(z_i, f_\theta\right)$$

where $f_\theta$ is the model to be trained, $D^{\text{train}} = \{z_i\}_{i=1}^{n}$ is the training dataset consisting of $n$ pairs of an input $x_i$ and a label $y_i$, i.e., $z_i = (x_i, y_i)$, and the total loss $L$ is the sum of the losses $l$ of each data point. The influence functions are defined with a perturbation $\epsilon$ for the loss of each data point $z_j$:

$$\hat{\theta}_{\epsilon,j} = \underset{\theta}{\operatorname{argmin}} \, L_{\epsilon,j}\left(D^{\text{train}}, f_\theta\right) = \underset{\theta}{\operatorname{argmin}} \, \frac{1}{n} \sum_{i=1}^{n} l\left(z_i, f_\theta\right) + \epsilon l\left(z_j, f_\theta\right),$$

which is considered as a shift of the probability that $z_j$ is sampled from the data distribution. The influence of the data $z_j$ on the model parameter $\hat{\theta}$ is defined as its increase rate with respect to this perturbation:

$$I^{\text{param}}(j) \overset{\text{def}}{=} \left. \frac{d\hat{\theta}_{\epsilon,j}}{d\epsilon} \right|_{\epsilon=0} = \left. -H^{-1} \frac{\partial l\left(z_j, f_\theta\right)}{\partial \theta} \right|_{\theta=\hat{\theta}} \tag{1}$$

where the Hessian is given by $H = \partial_\theta \partial_\theta L|_{\theta=\hat{\theta}}$. The influence on a differentiable function of $\theta$ is defined similarly, and calculated with the chain rule. For example, the influence on the loss of a test data $z_{\text{test}}$, which is a measure of the model's performance, is calculated to be

$$I^{\text{perf}}(z_{\text{test}}, j) \overset{\text{def}}{=} \left. \frac{dl\left(z_{\text{test}}, f_{\hat{\theta}_{\epsilon,z_j}}\right)}{d\epsilon} \right|_{\epsilon=0} = \left. \frac{dl\left(z_{\text{test}}, f_\theta\right)}{d\theta} \right|_{\theta=\hat{\theta}} \cdot I^{\text{param}}(j).$$

See Appendix A.1 for the derivation of equation 1. An underlying assumption is that the Hessian matrix is invertible, which is not always the case. Typically, in over-parameterized networks, the loss function often has non-unique minima with flat directions around them. In this paper, we examine how the definition of influence functions extends to cases with non-invertible Hessian matrix.

**Supervised meta-learning**. In this paper, we utilize TLXML for supervised meta-learning explanations (see for a review Hospedales et al. (2021)). In a typical supervised meta-learning setup, a task $\mathcal{T}$ is defined as a pair $\left(\mathcal{D}^{\mathcal{T}}, \mathcal{L}^{\mathcal{T}}\right)$ where $\mathcal{D}^{\mathcal{T}}$ represents the dataset and $\mathcal{L}^{\mathcal{T}}$ is the associated loss function for the supervised learning task. The occurrence of each task follows a distribution $\mathcal{T} \sim p\left(\mathcal{T}\right)$. An adaptation algorithm $\mathcal{A}$ takes as input a task $\mathcal{T}$ and meta-parameters $\omega$, and outputs

the weights $\hat{\theta}$ of the model $f_\theta$. The learning objective of a meta-learner is stated as the optimization of $\omega$ with respect to the test loss averaged over the task distribution:

$$\hat{\omega} = \operatorname*{argmin}_{\omega} \underset{\mathcal{T} \sim p(T)}{E} \left[ \mathcal{L}^{\mathcal{T}}(\mathcal{D}^{\mathcal{T},\text{test}}, f_{\hat{\theta}^{\mathcal{T}}}) \right] \quad \text{with } \hat{\theta}^{\mathcal{T}} = \mathcal{A}(\mathcal{T}, \omega)$$

A more practical formulation avoids explicitly using the notion of task distribution. Instead, it uses sampled tasks as building blocks. Task samples are divided into a taskset for training meta-parameters(source taskset) $D^{\text{src}} = \{\mathcal{T}^{\text{src(i)}}\}_{i=1}^{M}$ and a taskset for testing them(target taskset) $D^{\text{trg}} = \{\mathcal{T}^{\text{trg(i)}}\}_{i=1}^{M'}$ and the learning objective is framed as empirical risk minimization:

$$\hat{\omega} = \operatorname*{argmin}_{\omega} \frac{1}{M} \sum_{i=1}^{M} \mathcal{L}^{\text{src(i)}}(\mathcal{D}^{\text{src(i)test}}, f_{\hat{\theta}^i}) \quad \text{with } \hat{\theta}^i = \mathcal{A}(\mathcal{D}^{\text{src(i)train}}, \mathcal{L}^{\text{src(i)}}, \omega) \tag{2}$$

One performance metric in meta-testing is the test loss $\mathcal{L}^{\text{trg(i)}}(\mathcal{D}^{\text{trg(i)test}}, f_{\hat{\theta}^i})$ where $\hat{\theta}^i = \mathcal{A}(\mathcal{D}^{\text{trg(i)train}}, \mathcal{L}^{\text{trg(i)}}, \hat{\omega})$. In our experiments, we use and Prototypical Network (Snell et al., 2017) (see Appendix B.3.2) as commonly used examples of meta-learning algorithms. For MAML, the initial values $\theta_0$ of the network weights serve as the meta-parameters, and $\mathcal{A}$ represents a one-step gradient descent update of the weights with a fixed learning rate $\alpha$: $\hat{\theta}^i = \mathcal{A}(\mathcal{D}^{(i)\text{train}}, \mathcal{L}^i, \theta_0) = \theta_0 - \alpha \, \partial_\theta \mathcal{L}^i(\mathcal{D}^{(i)\text{train}}, f_\theta)\big|_{\theta=\theta_0}$.

For Prototypical Network, the meta-parameters are the weights of a feature extractor $f_\theta$ and the adaptation algorithm $\mathcal{A}$ is independent of the loss function. It passes the weights $\theta$ without any modification and simply calculates the feature centroid $c_k$ for each class $k$ based on the subset $S_k \in \mathcal{D}$: $\theta = \mathcal{A}_\theta(\mathcal{D}^{(i)\text{train}}, \mathcal{L}^i, \theta)$, and $c_k = \mathcal{A}_k(\mathcal{D}^{(i)\text{train}}, \mathcal{L}^i, \theta) = (1/|S_k^{(i)}|) \sum_{(x,y) \in S_k^{(i)}} f_\theta(x)$. The loss function $\mathcal{L}$ is not used in the adaptation of the Prototypical Network; it is only used in the outer loop. With $d$ being a distance measure (e.g. Euclidean distance), the class prediction for the data point $x$ in the test set $\mathcal{D}^{(i)\text{test}}$ is give by: $P_\theta(i|x) = \frac{\exp d(f_\theta(x), c_i)}{\sum_k \exp d(f_\theta(x), c_k)}$.

# 4 PROPOSED METHOD

## 4.1 TASK-LEVEL INFLUENCE FUNCTIONS

We now describe our method. TLXML measures the influence of training tasks on the adaptation and inference processes in meta-learning. To measure the influence of a training task $\mathcal{T}^j$ on the model's behaviors, we consider the task-level perturbation of the empirical risk defined in Equation 2:

$$\hat{\omega}_\epsilon^j = \operatorname*{arg\,min}_{\omega} \frac{1}{M} \sum_{i=1}^{M} \mathcal{L}^i(\mathcal{D}^{(i)\text{test}}, f_{\hat{\theta}^i}) + \epsilon \mathcal{L}^j(\mathcal{D}^{(j)\text{test}}, f_{\hat{\theta}^j}) \quad \text{with } \hat{\theta}^i = \mathcal{A}(\mathcal{D}^{(i)\text{train}}, \mathcal{L}^i, \omega). \tag{3}$$

The influence on $\hat{\omega}$ is measured as:

$$I^{\text{meta}}(j) \stackrel{\text{def}}{=} \left. \frac{d\hat{\omega}_\epsilon^j}{d\epsilon} \right|_{\epsilon=0} = - H^{-1} \left. \frac{\partial \mathcal{L}^j(\mathcal{D}^{(j)\text{test}}, f_{\hat{\theta}^j})}{\partial \omega} \right|_{\omega=\hat{\omega}} \tag{4}$$

where the Hessian matrix, $H$, is defined as follows:

$$H = \frac{1}{M} \sum_{i=1}^{M} \left. \frac{\partial^2 \mathcal{L}^i(\mathcal{D}^{(i)\text{test}}, f_{\hat{\theta}^i})}{\partial \omega \partial \omega} \right|_{\omega=\hat{\omega}}. \tag{5}$$

See Appendix A.1 for the derivation of Equation 4. The model's behavior is affected by the perturbation through the adapted parameters $\hat{\theta}_\epsilon^{ij} \equiv \mathcal{A}(\mathcal{D}^{(i)\text{train}}, \mathcal{L}^i, \hat{\omega}_\epsilon^j)$. Therefore, the influence of the training task $\mathcal{T}^j$ at the original parameter values $\hat{\theta}^i = \mathcal{A}(\mathcal{D}^{(i)\text{train}}, \mathcal{L}^i, \hat{\omega})$ is measured by:

$$I^{\text{adpt}}(i, j) \stackrel{\text{def}}{=} \left. \frac{d\hat{\theta}_\epsilon^{ij}}{d\epsilon} \right|_{\epsilon=0} = \left. \frac{\partial \mathcal{A}\left(\mathcal{D}^{(i)\text{train}}, \mathcal{L}^i, \omega\right)}{\partial \omega} \right|_{\omega=\hat{\omega}} I^{\text{meta}}(j) \tag{6}$$

Consequently, the influence of the training task $\mathcal{T}^j$ on the loss of a test task $\mathcal{T}^i$ is measured by:

$$I^{\text{perf}}(i,j) \stackrel{\text{def}}{=} \left. \frac{d\mathcal{L}^i \left( \mathcal{D}^{(i)\text{test}}, f_{\hat{\theta}_\epsilon^{ij}} \right)}{d\epsilon} \right|_{\epsilon=0} = \left. \frac{\partial \mathcal{L}^i \left( \mathcal{D}^{(i)\text{test}}, f_\theta \right)}{\partial \theta} \right|_{\theta=\hat{\theta}^{(i)}} I^{\text{adpt}}(i,j) \tag{7}$$

Note that training tasks are used only for evaluating $I^{\text{meta}}$. This means that we can obtain other explanation data without requiring accessing the original raw data. By retaining only the calculated $I^{\text{meta}}$ from the meta-learning process, we can mitigate storage concerns, making this approach suitable for devices with limited storage capacity.

**Task grouping**. In some cases, the abstraction of task-level explanation is not enough, and the explanation based on the task groups is more appropriate. This happens when the tasks used in the training are similar to each other for human intuition. For example, when an image recognition model is trained with task augmentation(see Ni et al. (2021a) for terminologies of data augmentation for meta-learning), e.g., flipping, rotating, or distorting the images in original tasks. In this case, the influence of each deformed task is not of interest; rather, the influence of a group consisting of tasks made from a single original task is of interest.

We extend the definition of influence functions to the task-group level by considering a common perturbation $\epsilon$ in the losses of tasks within a task-group $\mathcal{G}^J = \{\mathcal{T}^{j_0}, \mathcal{T}^{j_1}, \cdots\}$:

$$\hat{\omega}_\epsilon^J = \arg\min_\omega \frac{1}{M} \sum_{i=1}^M \mathcal{L}^i(\mathcal{D}^{(i)\text{test}}, f_{\hat{\theta}^i}) + \epsilon \sum_{\mathcal{T}^j \in \mathcal{G}^J} \mathcal{L}^j(\mathcal{D}^{(j)\text{test}}, f_{\hat{\theta}^j}) \quad \text{with } \hat{\theta}^i = \mathcal{A}(\mathcal{D}^{(i)\text{train}}, \mathcal{L}^i, \omega), \tag{8}$$

which modifies the influence function in Equation 4 as:

$$I^{\text{meta}}(J) \stackrel{\text{def}}{=} \left. \frac{d\hat{\omega}_\epsilon^J}{d\epsilon} \right|_{\epsilon=0} = \sum_{\mathcal{T}^j \in \mathcal{G}^J} I^{\text{meta}}(j) \tag{9}$$

Equation 6 and Equation 7 are only affected by replacing the task index $j$ with the task-group index $J$. See Appendix A.1 for the derivation of Equation 9.

### 4.2 HESSIAN APPROXIMATION METHOD VIA THE GAUSS-NEWTON MATRIX

TLXML faces computational barriers when applied to large models, particularly (as also noted in Alaa & Van Der Schaar (2020)) due to the computational costs of handling the Hessian in Equation 5. Although the Hessian is defined as the second-order tensor of the meta parameters, third-order tensors appear during its computation, resulting in a computational cost of at least $\mathcal{O}(pq^2)$ for a model with $p$ weights and $q$ meta-parameters. This is due to the bi-level structure of meta-learning (see Appendix A.2 for details). Furthermore, as is common in matrix inversion issues, inverting the Hessian incurs a computational cost of $\mathcal{O}(p^3)$.

To address this limitation in the expressiveness of meta-parameters, we propose using the Gauss-Newton matrix to approximate the Hessian matrix. For specific loss functions, e.g., mean squared error and cross-entropy, the Hessian can be decomposed into a sum of outer products of two vectors along with terms containing second-order derivatives. In this work, we only focus on the case of cross-entropy with the softmax function, which leads to

$$H = \frac{\partial^2}{\partial \omega \partial \omega} L = \sum_{njk} \sigma_k(\boldsymbol{y}_n) \left( \delta_{kj} - \sigma_j(\boldsymbol{y}_n) \right) \frac{\partial y_{nk}}{\partial \omega} \frac{\partial y_{nj}}{\partial \omega} - \sum_{njk} t_{nj} \left( \delta_{jk} - \sigma_k(\boldsymbol{y}_n) \right) \frac{\partial^2 y_{nk}}{\partial \omega \partial \omega} \tag{10}$$

where $y$ is the output of the last layer, $\sigma$ is the softmax function, $t$ is the one-hot vector of the target label, $j, k$ are class indices, and $n$ is the index specifying the combination of the input task and its corresponding data. The first term of Equation 10 is known as the Gauss-Newton matrix or the Fisher information metric. The coefficients of the second-order derivatives in the second term are the differences between predictions and target labels, and their sum equals zero. Since the second-order derivatives give rise to third-order tensors, we should discuss when these derivatives are uncorrelated with the coefficients, enabling the Hessian to be approximated by the first term.

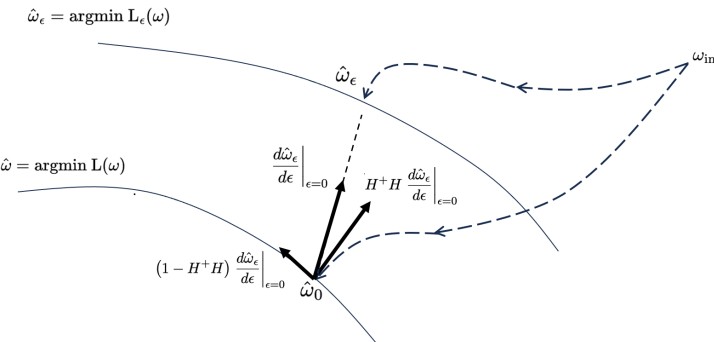

Figure 2: Diagram of the projected influence function, which measures the influence of a training task on the meta-parameters with the Hessian flat directions projected out.

Approximating the Hessian using the Gauss-Newton matrix is well-established in supervised learning (see for example Botev et al. (2017)). The difference in our case is the variables: in supervised learning, $L$ and $\mathbf{y}$ depend on the model's weights $\theta$, whereas in our case, they depend on the meta-parameters $\omega$. Putting some basic facts about cross-entropy (see A.3), we conclude that if there exists a distribution $P(X|\omega^*)$ that is well approximated by the training taskset, and the training result $\hat{\omega}$ is close to $\omega^*$, Equation 10 is dominated by its first term:

$$
\frac{\partial^2}{\partial \omega_\mu \partial \omega_\nu} L \bigg|_{\omega=\hat{\omega}} \sim \sum_{njk} \sigma_k(\boldsymbol{y}_n)(\delta_{kj} - \sigma_j(\boldsymbol{y}_n)) \frac{\partial y_{nk}}{\partial \omega_\mu} \frac{\partial y_{nj}}{\partial \omega_\nu} \bigg|_{\theta=\hat{\omega}}
$$
$$
= \sum_{nj} (\mathbf{V})_{\mu(nj)} (\mathbf{V}^{\mathrm{T}})_{(nj)\nu} \tag{11}
$$

where we introduce the factorized form of expression using the matrix $\mathbf{V}$ with the parameter index $\mu$ as a row index and $(nj)$ as a column index. In the case of a model with $p$ meta-parameters, a taskset with $M$ tasks, $n$ data point per task, and $c$ target classes, the size of $V$ is $p \times cnM$. For the approximated Hessian to be positive definite, $p$ must be as small as $cnM$, which limits the expressiveness of $\omega$. even if this condition is met, positive definiteness is not guaranteed. Instead, we extend the definition of influence functions to allow flat directions in the Hessian.

### 4.3 Extending Influence Functions with Flat Directions in the Hessian

Fig.2 depicts a generic case where flat directions of the Hessian appear in the parameter space. When the number of parameters is sufficiently large, the points that satisfy the minimization condition, $\hat{\omega} = \operatorname{argmin} L(\omega)$, form a hyper-surface, resulting in flat directions of the Hessian. The same holds true for the perturbed loss $L_\epsilon(\omega)$ used for defining the influence functions. The position along the flat directions resulting from minimization depends on the initial conditions and the learning algorithm. We do not explore this dependency in this paper. Instead, we employ a geometric definition of influence functions. Specifically, we take the partial inverse $H^+$ of $H$ in the subspace perpendicular to the flat directions, known as the pseudo-inverse matrix. With this approach, we modify the definitions of influence functions as:

$$
I^{\mathrm{meta}}(j) \stackrel{\mathrm{def}}{=} H^+ H \left. \frac{d\hat{\omega}_\epsilon^j}{d\epsilon} \right|_{\epsilon=0} = - H^+ \frac{\partial \mathcal{L}^j(D^{(j)\mathrm{test}}, f_{\hat{\theta}^j})}{\partial \omega} \bigg|_{\omega=\hat{\omega}}. \tag{12}
$$

See Appendix A.1 for the derivation of the second equation. $H^+H$ represents the projection that drops the flat directions. When the Hessian is approximated by the Fisher information metric, those are viewed as the direction in which the data distribution remains unchanged. The influence function, projected by $H^+H$, expresses the sensitivity of $\hat{\omega}$ in the steepest direction of the distribution change.

Note that $H^+$ can be computed without diagonalizing $H = VV^T$. Instead, this can be achieved by first finding an orthogonal matrix $O$ that diagonalizes $V^TV$, i.e., $V^TV = O\Lambda O^T$. Then, $VV^T$ is

implemented as a sequence of unnormalized projections in the direction of the column vectors of $VO = [v_1, v_2, \cdots]$, i.e., $VV^T = \sum v_i v_i^T$. The norms of the projections with non-vanishing vectors are adjusted accordingly, i.e., $v_i v_i^T \rightarrow v_i v_i^T / |v_i|^4$ for $|v_i| > 0$. Note that this method is feasible if the number of columns in $V$ is small. Although the Gauss-Newton matrix approximation avoids the $\mathcal{O}(pq^2)$ computational cost, the size of $V$ remains large, posing significant storage/memory cost. [1] Fortunately, the number of independent columns in $V$ is expected to be small, as most columns are dropped as zero vectors during orthogonalization. This number corresponds to the non-flat directions, representint the constraints imposed by the loss minimization condition. Typically, this is at most the number of training tasks. Exceptional cases arise when the model has weak adaptation ability, yet some training tasks are perfectly fitted $\mathcal{L}^i(\mathcal{D}^{(i)\text{test}}, \hat{\omega}) = 0$. In such cases, the sum of perfectly fitted data across tasks can increase the number of constraints on $\hat{\omega}$, thereby reducing the number of flat directions. In all other cases, where task losses take non-zero values, the number of the constraints on $\hat{\omega}$ is at most equal to the number of tasks.

## 5 EXPERIMENTS

The definition and approximation of influence functions in TLXML rely on certain assumptions: equation 4 is defined with the meta-parameters $\hat{\omega}$ at the exact optimum point, and the approximation in equation 11 is based on the assumption that the number of training samples is sufficiently large and $\hat{\omega}$ fits well to the target distribution. The experiments in this section aim to investigate whether the outputs of TLXML can be used to measure the influence of a training task on the meta-parameters numerically estimated by stochastic gradient descent. We empirically investigate two fundamental properties:

**Property 1**: If the network memorizes a training task, its influence on a test task with similar characteristics should be higher than the influence of other training tasks. We investigate this to validate the exact formulas in Equation 4, 6, and 7. For this validation, pairs of similar training and test tasks are created by setting the test task as identical to one of the training tasks.

**Property 2**: If the network encodes information about the training task distributions, tasks sampled from distributions similar to the test task should have a higher measured influence than tasks sampled from other distributions. We examine this to validate the approximation methods in Equation 11 and 12, as encoding task distribution information requires a sufficiently large network.

**Setup**. We use MAML as a meta-learning algorithm. We conduct the experiments with 5-ways-5-shots problem, taken from MiniImagenet (Vinyals et al., 2016) dataset. The implementation is based on the meta-learning library lean2learn (Arnold et al., 2020) and PyTorch's automatic differentiation package. The meta-parameters are trained with Adam. The meta batch size is set to 32, meaning the meta-parameters are updated after accumulating the gradients with 32 randomly selected tasks. We also conduct experiments with Prototypical Network and Omniglot dataset, with results in Appendix B.

### 5.1 DISTINCTION OF TASKS

To validate Property 1, we train a two-layer fully-connected network with widths of 32 and 5 (1285 parameters) with 1000 meta-batches. We use 128 training tasks to facilitate training progress with this small network. The model's input is a 32-dimensional feature vector extracted from the images by using Bag-of-Visual-Words (Csurka et al., 2004) with SIFT descriptor (Lowe, 1999; 2004) and k-means clustering.

First, we use the training tasks as test tasks to assess whether the network can distinguish the training task that is identical to each test task from other training tasks. Figure 3 shows the results, where Figure 3a illustrates a successful case, while Figure 3b reveals that in some tests, the training tasks that of the exact match are not ranked first, although they generally rank high in most cases. This instability might be because of the non-convexity of the training loss. In our case, a large fraction of the 1285 Hessian eigenvalues are near zero, and 92 eigenvalues are negative, violating the underlying

---

[1]For example, if the network has 100k parameters, the training taskset has 1000 tasks, and each task is a 5-way-5-shot problem, then the number of matrix elements is $p \times cnM = 10^5 \times (5 \times 25 \times 1000) \sim 10^{10}$, which makes it challenging to use floating-point numbers of 16, 32, or 64-bit precision.

Table 1: Effect of Hessian pruning. "# eigenvalues" denotes the number of largest eigenvalues treated as non-zero when computing the pseudo-inverse, with the rest set to zero (the 1st row is the original Hessian). The 2nd row corresponds to the pruned Hessian, where all 92 negative eigenvalues are set to zero. The 2nd column shows the self-ranks in the tests without degradation. The 2nd column lists self-ranks for tests without degradation, and the remaining columns show correlation coefficients between the two degradation parameters and the self-ranks or self-scores. Values are reported as means and standard deviations across 128 tasks.

| # eigenvalues | selfrank(avg±std) | correlation with degradation(avg±std) | | | |
| --- | --- | --- | --- | --- | --- |
| | | alpha/rank | alpha/score | ratio/rank | ratio/score |
| 1285 | 12.6±18.9 | 0.51±0.32 | -0.41±0.29 | 0.36±0.32 | -0.11±0.31 |
| 1193 | 0.0±0.0 | 0.69±0.21 | -0.69±0.22 | 0.46±0.30 | -0.15±0.34 |
| 512 | 0.0±0.0 | 0.71±0.12 | -0.96±0.06 | 0.63±0.09 | -0.89±0.13 |
| 256 | 0.0±0.0 | 0.71±0.11 | -0.95±0.08 | 0.62±0.11 | -0.88±0.14 |
| 128 | 0.0±0.0 | 0.72±0.10 | -0.94±0.04 | 0.63±0.10 | -0.86±0.15 |
| 64 | 0.0±0.0 | 0.71±0.12 | -0.92±0.06 | 0.66±0.12 | -0.77±0.20 |
| 32 | 0.0±0.2 | 0.72±0.16 | -0.85±0.12 | 0.68±0.14 | -0.68±0.22 |
| 16 | 2.0±3.2 | 0.67±0.22 | -0.69±0.20 | 0.61±0.22 | -0.46±0.27 |
| 8 | 8.6 ±9.1 | 0.55±0.26 | -0.53±0.23 | 0.47±0.27 | -0.29±0.31 |

assumption of Equation 4 (See AppendixB.2.1 for details). A potential solution is to treat small and negative eigenvalues as zero and use the pseudo-inverse of Hessian (instead of the inverse), as in Equation 12. The second column of Table 1 shows the effect of this solution, where the training tasks of the exact match are consistently ranked first.

To evaluate the performance in the scoring of training tasks that are not identical but similar to the test task, we degrade the test tasks by darkening some of the images in the task and examine whether the ranks and scores of the originally identical training tasks get worse as the similarity decreases. Figure 4 shows examples where the ranks and scores tend to get worse as the amount of darkness or the proportion of dark images increases. These examples include a small amount of Hessian pruning. We investigate the effects of pruning for these tests. The third to sixth columns of Table 1 show the correlation coefficients between the degradation parameters and the ranks and the scores of the originally identical training tasks. We can see that pruning increases the correlation values.

## 5.2 DISTINCTION OF TASK DISTRIBUTIONS

To validate Property 2, we construct a training set of tasks sampled from different distributions by mixing 128 tasks made up of noise images with 896 training tasks from MiniImagenet (in total 1024 tasks). We train a CNN with four convolutional layers and one fully connected layer (121,093 parameters). To see the effect of generalization, we apply task augmentation and weight decay during training. For task augmentation, we rotate images in each training task. Specifically, for each training task, we randomly select 2, 4, 8, or 16 angles between $-\pi/2$ and $+\pi/2$ and we create new tasks

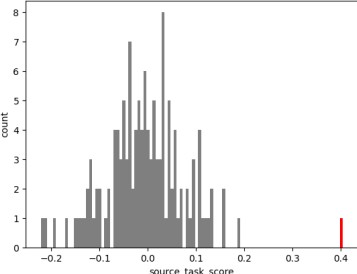

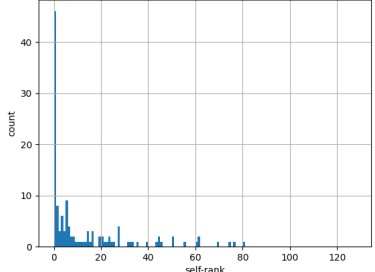

(a) Example of training task score distribution in a single test. Highlighted is the training task identical to the test task.

(b) Distribution of the self-ranks across the 128 tests. We define the self-rank as the rank of the training task identical to the test task.

Figure 3: Test with training tasks. As shown in (a), the task most similar to the test task is successfully separated from the others by using TLXML.

by rotating all images with those angles. We take 128 test tasks from the MiniImagenet test taskset. For each training condition, we evaluate the influences of the 1024 training tasks on the 128 test loss values based on Equation 6, and 7 with the projected influence on the meta-parameters (Equation 12) and the Gauss-Newton matrix approximation of the Hessian (Equation 10). For the cases where we employ task augmentation, we use the technique to define group influence equation 9.

Figure 5 shows the results of the experiments. According to the figure, the distributions of the scores for both the regular training tasks and the noise image tasks overlap, showing no clear distinction in each test. However, when analyzing the statistics over the 128 tests, the order of mean scores shows a noticeable difference between the two distributions. We define the mean values of the scores for the regular and noise image tasks to be in *proper order* if the mean score of the regular tasks is greater than that of the noise image tasks. Table 2 shows the number of tests exhibiting the proper order and their p-values from binomial tests across various training settings. We observe a trend, that is, when the model fits well to the training tasks, the scores of regular and noise tasks are in the opposite of proper order. Also, as we enhance the model's generalization via data augmentation (image rotation or weight decay), the scores align in the proper order. This observation can be interpreted as reflecting a specific characteristic of model behavior. During overfitting, only a few training tasks that are similar to the test task provide useful information, while the majority of regular training tasks become detrimental. Conversely, when generalization occurs, the model effectively encodes information from the training task distributions, making tasks from distributions similar to the test task generally beneficial. In contrast, the noise image tasks have a neutral impact on the test performance, as they do not contribute a non-vanishing gradient on average. Given that the p-values are statistically significant in both overfitting and generalization scenarios, we conclude that TLXML can effectively analyze these phenomena. See Appendix B.3 for experimental details.

## 6 DISCUSSION AND CONCLUSION

This paper presented TLXML, a method for quantifying the impact of training tasks on the meta-parameters, adapted network weights, and inference outcomes. Through experiments with a small network and two meta-learners, we observed that TLXML effectively serves as a similarity measure between training and test tasks. In Figure 6, this observation is qualitatively explored. The figure shows image samples from similar (Figure 6b and Figure 6c) and dissimilar (Figure 6d) training tasks to a test task (Figure 6a) that received high and low scores, respectively.

Additionally, we proposed a modification of the influence functions to address cases where the Hessian has flat directions. We also explored how the computational cost of managing the Hessian for large networks can be reduced by using the quadratic form with the Gauss-Newton matrix. In our experiments with a middle-sized CNN, we observed that the approximated influence scores were effective in distinguishing between training task distributions in several cases.

One limitation of our work is that the definitions of the influence functions rely on the assumption of a local minimum of the loss function. Future work could aim to design influence functions that account for early stopping techniques. Additionally, our discussion of the approximation method

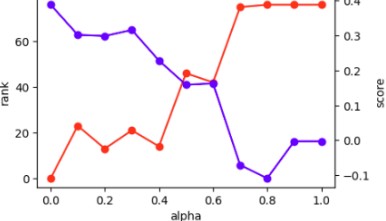 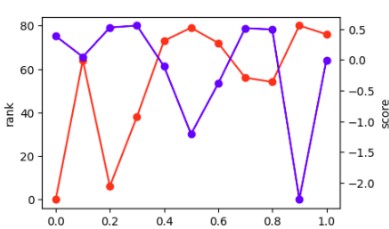

(a) Example of the effect of increasing $\alpha$. $\alpha{=}1$ means all the images in the task are dark images.

(b) Example of the effect of increasing the ratio. ratio=1 means all dark images are completely black.

Figure 4: Test with degraded training tasks. The parameters $\alpha$ and ratio specify the darkness of images, and the proportion of the dark images, respectively. The red and blue lines represent ranks and scores, respectively. Both examples were performed with the Hessian pruned to retain only the 1193 most significant eigenvalues.

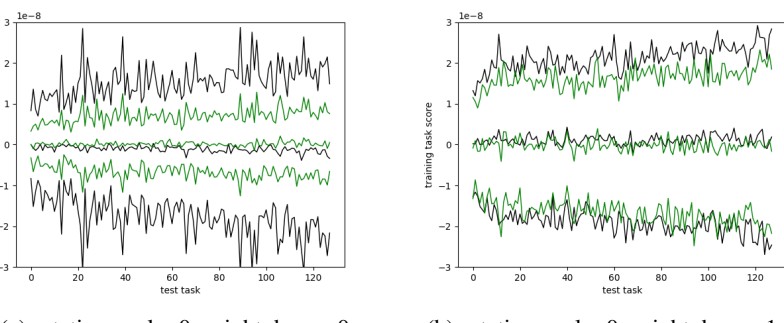

(a) rotation angle: 0 weight decay: 0    (b) rotation angle: 0 weight decay: 1e-3

Figure 5: Experiment with mixed training tasksets: examples of a score distribution. Lines show mean values and the standard deviations of the scores of training tasks. Black and green lines represent regular and noise image tasks, respectively, with 128 tests arranged horizontally by ascending test error.

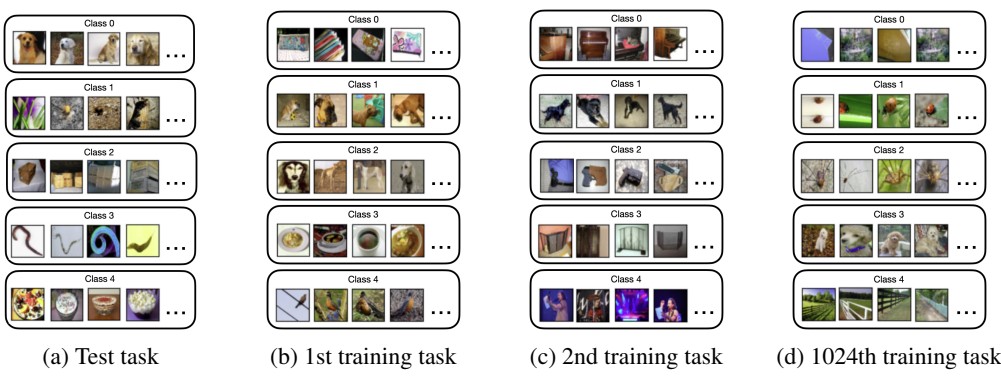

(a) Test task        (b) 1st training task        (c) 2nd training task        (d) 1024th training task

Figure 6: A test task (accuracy=0.76)(a) and three training tasks ranked 1st(b), 2nd(c), and 1024th(d) by TLXML. According to this figure, in the test task, class 0 is similar to classes 1 and 2 from the first training task, while class 4 is similar to class 3. Similarly, class 0 in the test task resembles class 3 of the 1024th training task. Additionally, class 2 in the test task is similar to class 0 of both the first and second training tasks, and class 4 is again similar to class 3 of the first training task. Interestingly, class 1 in the test task is similar to classes 0, 1, and 2 of the 1024th training task, suggesting that models trained on these classes may struggle to classify class 1 correctly, making the 1024th training task ineffective for accurate explanations.

focuses solely on classification tasks with cross-entropy loss in meta-learning. Nevertheless, we expect that future work focusing on extending TLXML to other task types, e.g., regression and reinforcement learning, should be straightforward, similar to its application in supervised learning.

Table 2: Experiments with a CNN trained with MAML on MiniImagenet dataset, combined with 128 noise image tasks. The 128 test tasks were drawn from the pure MiniImagenet test taskset. The number of tests in which the mean training task scores are in proper order is presented in the 6th column. The p values of the binomial test are also listed. The standard deviation $\sigma$ of the binomial distribution under the null hypothesis is calculated as $\sqrt{128 \times 0.5^2} \sim 5.66$.

| training parameter | | | accuracy | | test in proper order | |
|---|---|---|---|---|---|---|
| # rotation angles | weight decay | iteration | test | train | count | p-value |
| 1(0 radian) | 0 | 12000 | 0.43 | 1.0 | 10 $(-9.5\sigma)$ | $1.5 \times 10^{-24}$ |
| 2 | 0 | 40000 | 0.40 | 0.99 | 42 $(-3.9\sigma)$ | $1.3 \times 10^{-4}$ |
| 4 | 0 | 40000 | 0.39 | 0.95 | 57 $(-1.2\sigma)$ | 0.25 |
| 8 | 0 | 40000 | 0.48 | 0.86 | 90 $(+4.6\sigma)$ | $4.9 \times 10^{-6}$ |
| 16 | 0 | 40000 | 0.41 | 0.77 | 101 $(+6.5\sigma)$ | $3.0 \times 10^{-11}$ |
| 1(0 radian) | 0.00001 | 16000 | 0.41 | 1.00 | 16 $(-8.5\sigma)$ | $6.4 \times 10^{-19}$ |
| 1(0 radian) | 0.0001 | 20000 | 0.41 | 0.97 | 89 $(+4.4\sigma)$ | $1.2 \times 10^{-5}$ |
| 1(0 radian) | 0.001 | 20000 | 0.45 | 0.95 | 100 $(+6.4\sigma)$ | $1.1 \times 10^{-10}$ |

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

## TECHNICAL APPENDICES

## A  METHOD DETAILS

### A.1  IMPLICIT DIFFERENTIATION

The second equation in each of Equation 1, Equation 4, Equation 9, and Equation 12 is derived from the following property.

*Property 1*: If a vector parameter $\hat{\theta}_\epsilon$ is parametrized by a scalar parameter $\epsilon$ in such a way such the local maximum or the local minimum condition of a second-order differentiable function $L(\theta, \epsilon)$ is satisfied for each value of $\epsilon$, then the derivative of $\hat{\theta}_\epsilon$ with respect to $\epsilon$ satisfies:

$$\left. \frac{\partial^2 L(\theta, \epsilon)}{\partial\theta\partial\theta} \right|_{\theta=\hat{\theta}_\epsilon} \frac{d\hat{\theta}_\epsilon}{d\epsilon} = - \left. \frac{\partial L(\theta, \epsilon)}{\partial\theta\partial\epsilon} \right|_{\theta=\hat{\theta}_\epsilon}. \tag{13}$$

The proof is done almost trivially by differentiating the local maximum or minimum condition:

$$\left. \frac{\partial L(\theta, \epsilon)}{\partial\theta} \right|_{\theta=\hat{\theta}_\epsilon} = 0 \tag{14}$$

with respect to $\epsilon$ and apply the chain rule. Note that if the matrix $\partial\partial L$ in Equation 13 is invertible, we can solve the equation to obtain the $\epsilon$-derivative of $\hat{\theta}_\epsilon$. Note also that we do not assume $\hat{\theta}_\epsilon$ to be the unique solution of Equation 14 and Equation 13 is true for any parametrization of $\theta$ with $\epsilon$ that satisfies Equation 14.

### A.2  THIRD-ORDER TENSORS IN INFLUENCE FUNCTIONS

Here, we explain how the computational cost of $\mathcal{O}(pq^2)$ arises in evaluating the influence function in Equation 4. This is due to the third order tensors which appear in the intermediate process of evaluating the Hessian:

$$H = \frac{1}{M} \sum_{i=1}^{M} \frac{\partial^2 \mathcal{L}^i(\mathcal{D}^{(i)\text{test}}, f_{\hat{\theta}^i(\omega)})}{\partial\omega\partial\omega} \tag{15}$$

$$= \frac{1}{M} \sum_{i=1}^{M} \left[ \left( \frac{\partial\hat{\theta}^i(\omega)}{\partial\omega} \right)^{\mathrm{T}} \left. \frac{\partial^2 \mathcal{L}^i(\mathcal{D}^{(i)\text{test}}, f_\theta)}{\partial\theta\partial\theta} \right|_{\theta=\hat{\theta}^i(\omega)} \frac{\partial\hat{\theta}^i(\omega)}{\partial\omega} \right.$$
$$\left. + \left. \frac{\partial \mathcal{L}^i(\mathcal{D}^{(i)\text{test}}, f_\theta)}{\partial\theta} \right|_{\theta=\hat{\theta}^i(\omega)} \frac{\partial^2\hat{\theta}^i(\omega)}{\partial\omega\partial\omega} \right] \tag{16}$$

where $\hat{\theta}^i = \mathcal{A}(\mathcal{L}^{(i)\text{train}}, \mathcal{L}^i, \omega)$. Because $\hat{\theta}^i$ and $\omega$ is a $p$-dimensional $q$-dimensional and vector, the second-order derivative $\partial\partial\hat{\theta}^i$ in the last term is the third-order tensor of $pq^2$ elements. This tensor also appears in the evaluation of the second-order derivative of the network output with respect to $\omega$. In the case of MAML, this tensor is in the form of a third-order derivative:

$$\frac{\partial^2\hat{\theta}^i(\theta_0)}{\partial\theta_0\partial\theta_0} = \frac{\partial^2}{\partial\theta_0\partial\theta_0}\mathcal{A}(\mathcal{D}^{(i)\text{train}}, \mathcal{L}^i, \theta_0) = -\alpha\frac{\partial^3}{\partial\theta_0\partial\theta_0\partial\theta_0}\mathcal{L}^i\left(\mathcal{D}^{(i)\text{train}}, f_{\theta_0}\right).$$

### A.3  RELATIONS AMONG KL-DIVERGENCE, CROSS-ENTROPY, AND FISHER INFORMATION MATRIX

For the reader's convenience, we present basic facts related to the approximation method argued in section 4.2.

**Variant expressions of Hessian**  The cross-entropy $L$ between two probability distributions $P(X|\omega^*)$, $P(X|\omega)$ parametrized by $\omega^*$ and $\omega$, is equivalent to the Kullback–Leibler (KL) divergence up to a $\omega$-independent term:

$$D_{KL}\left(P(X|\omega^*), P(X|\omega)\right) = L\left(P(X|\omega^*), P(X|\omega)\right) + \int P(X|\omega^*)\log P(X|\omega^*)dX.$$

Therefore, the second-order derivatives of them with respect to $\omega$ are identical:

$$\frac{\partial^2}{\partial\omega\partial\omega}D_{KL} = \frac{\partial^2}{\partial\omega\partial\omega}L$$

Furthermore, considering the Taylor expansion of $D_{KL}$ with $\Delta\omega = \omega - \omega^*$

$$
\begin{aligned}
&D_{KL}\left(P\left(X|\omega^*\right), P\left(X|\omega\right)\right)\\
&\sim -\sum_\mu \left[\int \partial_\mu P\left(X|\omega^*\right)dX\right]\Delta\omega^\mu\\
&+ \frac{1}{2}\sum_{\mu\nu}\left[\int -\partial_\mu\partial_\nu P\left(X|\omega^*\right) + \frac{\partial_\mu P\left(X|\omega^*\right)\partial_\nu P\left(X|\omega^*\right)^2}{P\left(X|\omega^*\right)}dX\right]\Delta\omega^\mu\Delta\omega^\nu\\
&= \frac{1}{2}\sum_{\mu\nu}\mathrm{E}_{X\sim P(X|\omega^*)}\left[\partial_\mu\log P\left(X|\omega^*\right)\partial_\nu\log P\left(X|\omega^*\right)\right]\Delta\omega^\mu\Delta\omega^\nu.\\
&\equiv \frac{1}{2}\sum_{\mu\nu}g_{\mu\nu}\left(\omega^*\right)\Delta\omega^\mu\Delta\omega^\nu,
\end{aligned}
$$

we see that

$$\left.\frac{\partial^2}{\partial\omega_\mu\partial\omega_\nu}D_{KL}\right|_{\omega=\omega^*} = \left.\frac{\partial^2}{\partial\omega_\mu\partial\omega_\nu}L\right|_{\omega=\omega^*} = g_{\mu\nu}\left(\omega^*\right) \tag{17}$$

**Approximations by empirical sums**  Let us consider the case that $X = (x, c)$ is the pair of a network input $x_n$ and a class label $c_n$ and $P$ is the composition of soft-max function $\sigma$ and the network output $\boldsymbol{y}_n \equiv f_\omega(x_n)$. Assuming that sampled data accurately approximate the distributions, we obtain the expressions of the Fisher information metric in the form of an empirical sum:

$$
\begin{aligned}
g_{\mu\nu}\left(\omega^*\right) &= \mathrm{E}_{X\sim P(X|\omega^*)}\left[\partial_\mu\log P\left(X|\omega^*\right)\partial_\nu\log P\left(X|\omega^*\right)\right]\\
&= \mathrm{E}_{(c,x)\sim P_{\omega^*}(c|x)P(x)}\left[\partial_\mu\log\left(P_\omega\left(c|x\right)P(x)\right)\partial_\nu\log\left(P_\omega\left(c|x\right)P(x)\right)\right]\big|_{\omega=\omega^*}\\
&= \mathrm{E}_{(c,x)\sim P_{\omega^*}(c|x)P(x)}\left[\partial_\mu\log\left(P_\omega\left(c|x\right)\right)\partial_\nu\log\left(P_\omega\left(c|x\right)\right)\right]\big|_{\omega=\omega^*}\\
&\sim \sum_{ni}\sigma_i\left(\boldsymbol{y}_n\right)\left[\partial_\mu\log\left(\sigma_i\left(\boldsymbol{y}_n\right)\right)\partial_\nu\log\left(\sigma_i\left(\boldsymbol{y}_n\right)\right)\right]\Big|_{\omega=\omega^*}\\
&= \sum_{nijk}\sigma_i\left(\boldsymbol{y}_n\right)\left(\delta_{ik}-\sigma_k\left(\boldsymbol{y}_n\right)\right)\left(\delta_{ij}-\sigma_j\left(\boldsymbol{y}_n\right)\right)\frac{\partial y_{nk}}{\partial\omega_\mu}\frac{\partial y_{nj}}{\partial\omega_\nu}\Big|_{\omega=\omega^*}\\
&= \sum_{nkj}\sigma_k\left(\boldsymbol{y}_n\right)\left(\delta_{kj}-\sigma_j\left(\boldsymbol{y}_n\right)\right)\frac{\partial y_{nk}}{\partial\omega_\mu}\frac{\partial y_{nj}}{\partial\omega_\nu}\Big|_{\omega=\omega^*}.
\end{aligned}
$$

To express the Hessian in a similar way, we denote the target vector of a sample $x_n$ as $t_{ni}$. Then:

$$\frac{\partial^2}{\partial\omega\partial\omega} L = -\frac{\partial^2}{\partial\omega\partial\omega} \sum_c \int P_{\omega^*}(c|x) P(x) \log [P_\omega(c|x) P(x)] \, dx$$

$$\sim -\frac{\partial^2}{\partial\omega\partial\omega} \sum_{ni} t_{ni} \log [P_\omega(i|x_n) P(x_n)]$$

$$= -\frac{\partial^2}{\partial\omega\partial\omega} \sum_{ni} t_{ni} \log \sigma_i(\boldsymbol{y_n})$$

$$= -\frac{\partial}{\partial\omega} \sum_{nik} t_{ni} [\delta_{ik} - \sigma_k(\boldsymbol{y_n})] \frac{\partial y_{nk}}{\partial\omega}$$

$$= \sum_{nijk} t_{ni}\sigma_k(\boldsymbol{y_n}) (\delta_{kj} - \sigma_j(\boldsymbol{y_n})) \frac{\partial y_{nk}}{\partial\omega} \frac{\partial y_{nj}}{\partial\omega} - \sum_{nik} t_{ni} [\delta_{ik} - \sigma_i(\boldsymbol{y_n})] \frac{\partial^2 y_{nk}}{\partial\omega\partial\omega}$$

$$= \sum_{njk} \sigma_k(\boldsymbol{y_n}) (\delta_{kj} - \sigma_j(\boldsymbol{y_n})) \frac{\partial y_{nk}}{\partial\omega} \frac{\partial y_{nj}}{\partial\omega} - \sum_{nik} t_{ni} [\delta_{ik} - \sigma_i(\boldsymbol{y_n})] \frac{\partial^2 y_{nk}}{\partial\omega\partial\omega}$$

$$= g(\omega) - \sum_{nik} t_{ni} [\delta_{ik} - \sigma_i(\boldsymbol{y_n})] \frac{\partial^2 y_{nk}}{\partial\omega\partial\omega}. \tag{18}$$

The second to the last equation proves Equation 10. Thus we see that if the distributions $P(X|\omega^*)$ and $P(X|\omega)$ are accurately approximated by the training data samples and the model's outputs, respectively, the first term in Equation 10 becomes equivalent to the Fisher information metric evaluated at $\omega$. By comparing Equation 17 and Equation 18, we see that if $\omega$ is near to $\omega^*$, the second term of Equation 18 drops and the Hessian is well approximated by the Fisher information metric.

## B EXPERIMENTAL DETAILS

### B.1 COMPUTE RESOURCES

The experiments in Section 5 are carried out in multiple computing environments. We present the information on computation times in the case of an Intel Core i7-7567U CPU with a 3.50GHz clock and an NVIDIA GeForce RTX 2070 GPU.

For conducting the experiments in Section 5.1 with the small network with 1285 parameters, training the model with 1000 meta-bathes takes about $\approx 2$ hours, computing $I^{\mathrm{meta}}$ with the 128 training tasks without approximation takes about $\approx 20$ minutes, and after that, computing the influence scores $I^{\mathrm{perf}}$ for the 128 test tasks takes a few minutes.

For conducting the experiments in Section 5.2 with the CNN with 121,093 parameters, training the model with 40000 meta-batches takes about $\approx 1$ day, computing $I^{\mathrm{meta}}$ with the 1024 training tasks with the proposed approximation method takes about $\approx 15$ hours, and after that, computing the influence scores $I^{\mathrm{perf}}$ for the 128 test tasks takes about $\approx 2$ hours. If we employ data augmentation in training, we need to perform the summation over the augmented tasks in the computation of $I^{\mathrm{meta}}$, and its computational time scales proportionally with the amount of augmentation.

### B.2 DETAILS OF DISTINCTION OF TASKS (SECTION 5.1)

#### B.2.1 EFFECT OF PRUNING HESSIAN

In the experiments in Section 5.1 with MiniImagenet, we encountered negative eigenvalues of the Hessian. We provide some details here. The plot in Figure 7 shows 1285 eigenvalues arranged from the largest to the smallest. We can see that the non-zero eigenvalues are almost restricted to the first several hundred elements. We can also see 92 negative eigenvalues in the tail.

We also conducted experiments with the Omniglot dataset. We train a two-layer fully connected network with widths of 32 and 5 (1413 parameters) with 1000 meta-batches. We use 128 training tasks to facilitate training progress with this small network. The model's input is a 36-dimensional

Table 3: Effect of pruning the Hessian(Omniglot dataset). "# eigenvalues" means the number of the eigenvalues we choose from the largest and treat as non-zero in taking the pseudo inverse. We set the rest of the eigenvalues to zero. The first line is the case of the original Hessian, where we take the inverse without any additional manipulation. Actually, 428 of them are negative, and the second line is the case that we prune them. The second column presents the self-ranks in the tests without degradation. The rest of the columns present the correlation coefficients between the two degradation parameters and self-ranks or self-scores. All the values are given with the mean values and standard deviations across the 128 tests. There are cases in which increasing $\alpha$ does not change the ranks of the original training tasks. The numbers of those cases are shown in the brackets, and they are removed from the statistics because the correlation coefficients can not be defined for them.

| # eigenvalues | selfrank(avg±std) | correlation with degradation(avg±std) | | | |
|---|---|---|---|---|---|
| | | alpha/rank | alpha/score | ratio/rank | ratio/score |
| 1413 | 66.0 ±36.9 | 0.15 ±0.48(21) | 0.06 ±0.50 | 0.02 ±0.34 | 0.03 ±0.23 |
| 985 | 7.8 ±10.0 | 0.48 ±0.12(2) | -0.37 ±0.33 | 0.25 ±0.28 | 0.01 ±0.23 |
| 512 | 0.8 ±5.0 | 0.50 ±0.00(1) | -0.50 ±0.00 | 0.35 ±0.26 | -0.15 ±0.23 |
| 256 | 1.6 ±6.3 | 0.50 ±0.00(1) | -0.50 ±0.00 | 0.34 ±0.27 | -0.12 ±0.25 |
| 128 | 2.9 ±7.4 | 0.50 ±0.00(1) | -0.50 ±0.00 | 0.36 ±0.27 | -0.11 ±0.24 |
| 64 | 4.3 ±8.0 | 0.50 ±0.00(1) | -0.50 ±0.00 | 0.36 ±0.26 | -0.13 ±0.23 |
| 32 | 6.4 ±8.8 | 0.50 ±0.00(1) | -0.49 ±0.09 | 0.33 ±0.26 | -0.08 ±0.24 |
| 16 | 8.4 ±10.0 | 0.50 ±0.00(1) | -0.50 ±0.00 | 0.32 ±0.30 | -0.08 ±0.25 |
| 8 | 11.8 ±12.5 | 0.50 ±0.00(4) | -0.50 ±0.00 | 0.29 ±0.29 | -0.06 ±0.24 |

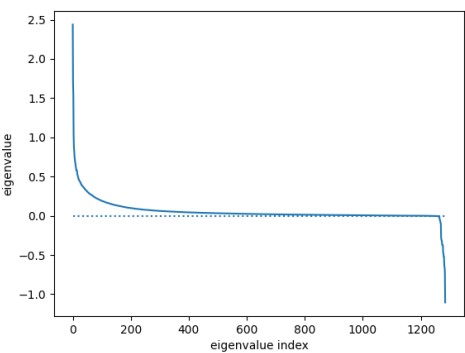

Figure 7: Eigenvalues of the Hessian before the pruning in 5.1

feature vector extracted from the image by applying 2-dimensional FFT and dropping the edges to get the 6x6 images at the center. In this case, we encountered 428 negative eigenvalues of the Hessian. The effects of pruning on the rank of test tasks chosen from the training task set are shown in the first column of Table 3. Tests with the degradation of those test tasks were also conducted, and the third to seventh columns of the table in Table 3 show the correlation coefficients between the degradation parameters and the ranks and the scores of the originally identical training tasks. Again, we see the values of correlation coefficients are increased by pruning to some extent, but those correlations are weaker than in the cases of MiniImagenet (Table 1). A possible reason for the weak correlation with is that the images of handwritten digits in Omniglot have small varieties of darkness and the trained model is more agnostic about the darkness of the images. A possible reason for the weak correlation with the ratio is that the tasks in Omniglot are easier to adapt to than the tasks in MiniImagenet.

### B.3 DETAILS OF DISTINCTION OF TASK DISTRIBUTIONS (SECTION 5.2)

#### B.3.1 MAML

**Implementation Details** As discussed in Section 4, approximating the Hessian using the quadratic form of the Gauss-Newton matrix is insufficient to fully address the storage or memory cost issue. Thus it is necessary to select a small number of independent vectors from the linear combinations of columns of $V$ in Equation 11. In the experiments of Section 5.2, 1024 independent vectors are

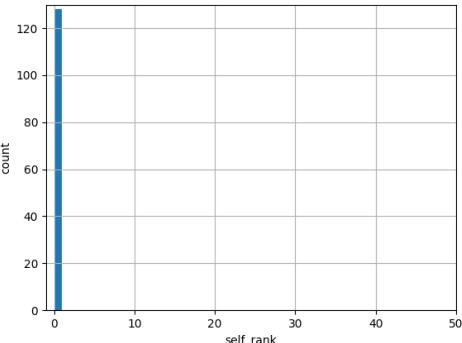

Figure 8: Distribution of self-ranks in test with the training tasks with CNN. Here we define the self-rank as the rank of the training task identical to the test task.

Table 4: Experiments with a CNN trained with MAML and MiniImagenet dataset combined with 128 noise image tasks. 128 test tasks were taken from the test taskset of pure MiniImagenet. The numbers of tests where mean and median training task scores are in proper order are listed in the last two columns. We say the mean or median scores of normal and noise image tasks are in proper order if the former is larger than the latter. The standard deviation $\sigma$ of the binomial distribution of the hypothesis is $\sqrt{128 \times 0.5^2} \sim 5.66$.

| training parameter | | | accuracy | | # test with proper score order | |
|---|---|---|---|---|---|---|
| # rotation angles | weight decay | iteration | test | train | mean | median |
| 1(0 radian) | 0 | 12000 | 0.43 | 1.00 | 10 $(-9.5\sigma)$ | 12 $(-9.2\sigma)$ |
| 2 | 0 | 40000 | 0.40 | 0.99 | 42 $(-3.9\sigma)$ | 25 $(-6.9\sigma)$ |
| 4 | 0 | 40000 | 0.39 | 0.95 | 57 $(-1.2\sigma)$ | 58 $(-1.1\sigma)$ |
| 6 | 0 | 40000 | 0.43 | 0.89 | 87 $(+4.1\sigma)$ | 81 $(+3.0\sigma)$ |
| 8 | 0 | 40000 | 0.48 | 0.86 | 90 $(+4.6\sigma)$ | 91 $(+4.8\sigma)$ |
| 10 | 0 | 40000 | 0.45 | 0.84 | 102 $(+6.7\sigma)$ | 97 $(+5.8\sigma)$ |
| 12 | 0 | 40000 | 0.49 | 0.84 | 83 $(+3.4\sigma)$ | 76 $(+2.1\sigma)$ |
| 14 | 0 | 40000 | 0.45 | 0.80 | 61 $(-0.5\sigma)$ | 53 $(-1.9\sigma)$ |
| 16 | 0 | 40000 | 0.41 | 0.77 | 101 $(+6.5\sigma)$ | 95 $(+5.5\sigma)$ |
| 1(0 radian) | 0.00001 | 16000 | 0.41 | 1.00 | 16 $(-8.5\sigma)$ | 15 $(-8.7\sigma)$ |
| 1(0 radian) | 0.0001 | 20000 | 0.44 | 0.97 | 89 $(+4.4\sigma)$ | 76 $(+2.1\sigma)$ |
| 1(0 radian) | 0.0005 | 20000 | 0.46 | 0.97 | 88 $(+4.2\sigma)$ | 90 $(+4.6\sigma)$ |
| 1(0 radian) | 0.001 | 20000 | 0.45 | 0.95 | 100 $(+6.4\sigma)$ | 91 $(+4.8\sigma)$ |

Table 5: Experiments with a CNN trained with MAML and Omniglot dataset combined with 128 noise image tasks. 128 test tasks were taken from the test taskset of pure Omniglot. See the caption of Table 4 for other notations.

| training parameter | | | | accuracy | | # test with proper score order | |
|---|---|---|---|---|---|---|---|
| # tasks | # normal tasks | # noise tasks | iteration | test | train | mean | median |
| 136 | 8 | 128 | 17000 | 0.531 | 1.00 | 58 $(-1.1\sigma)$ | 61 $(-0.53\sigma)$ |
| 144 | 16 | 128 | 9000 | 0.571 | 1.00 | 57 $(-1.2\sigma)$ | 63 $(-0.18\sigma)$ |
| 160 | 32 | 128 | 10000 | 0.631 | 1.00 | 65 $(+0.18\sigma)$ | 66 $(+0.35\sigma)$ |
| 192 | 64 | 128 | 18000 | 0.657 | 1.00 | 66 $(+0.35\sigma)$ | 75 $(+1.9\sigma)$ |
| 256 | 128 | 128 | 12000 | 0.747 | 1.00 | 51 $(-2.3\sigma)$ | 54 $(-1.8\sigma)$ |
| 512 | 384 | 128 | 20000 | 0.868 | 1.00 | 71 $(+1.2\sigma)$ | 75 $(+1.9\sigma)$ |
| 1024 | 896 | 128 | 18000 | 0.939 | 1.00 | 84 $(+3.5\sigma)$ | 80 $(+2.8\sigma)$ |
| 2048 | 1920 | 128 | 15000 | 0.965 | 1.00 | 73 $(+1.6\sigma)$ | 73 $(+1.6\sigma)$ |

Table 6: Experiments with a CNN trained with MAML and Omniglot dataset combined with noise image tasks at the mixing rate 7:1. 128 test tasks were taken from the test taskset of pure Omniglot. See the caption of Table 4 for other notations.

| training parameter | | | | accuracy | | # test with proper score order | |
|---|---|---|---|---|---|---|---|
| # tasks | # normal tasks | # noise tasks | iteration | test | train | mean | median |
| 128 | 112 | 16 | 20000 | 0.749 | 1.000 | 61$(-0.53\sigma)$ | 53$(-1.94\sigma)$ |
| 256 | 224 | 32 | 20000 | 0.829 | 1.000 | 61$(-0.53\sigma)$ | 65$(+0.18\sigma)$ |
| 512 | 448 | 64 | 20000 | 0.894 | 1.000 | 76$(+2.12\sigma)$ | 76$(+2.12\sigma)$ |
| 1024 | 896 | 128 | 18000 | 0.939 | 1.000 | 84$(+3.54\sigma)$ | 80$(+2.83\sigma)$ |

collected. At each step in the summation of Equation 11 over the training tasks, the columns of $V$ are orthogonalized, the vectors with the largest 1024 norms are kept, and the other vectors are dropped.

To evaluate if the buffer of vectors with the size 1024 has enough expressiveness, we trained the same CNN with 1024 regular tasks of MiniImagenet, approximated the Hessian using the same method, and conducted 128 tests with the tasks taken from the training taskset as in the case of 5.1. Figure 8 shows that the training task identical to each of the test tasks is perfectly distinguished by the influence score calculated using the aforementioned approximation.

**MiniImagenet** Table 4 shows the full list of the results of the experiments with mixed training task distributions. In the last two columns, we present the numbers of the tests where the two distributions are in proper order both in terms of mean values and median values. Again, we observe the tendency that when the model fits well to the training tasks, the scores of regular and noise image tasks are distributed in the opposite of proper order, and as we increase the generalization ability through data augmentation with image rotation or weight decay, they align in proper order.

**Omniglot** For Omniglot dataset, we trained a CNN with four-layer convolutional layers and one fully connected layer(111,261 parameters). Table 5 and 6 shows the results with the Ominglot dataset mixed with noise image training task. Ominglot tasks are known to be easier than MiniImagenet, therefore we can investigate the region of higher test accuracies. Note that no data augmentation nor use weight decay were necessary to achieve test accuracies above 0.9. Rather, we reduced the number of training tasks to lower the level of generalization. We examined two methods for reducing the number of training tasks: 1) reducing the number of normal tasks and 2) reducing the number of both normal and noise tasks with a fixed mixing rate. The tables once again show the tendency of the relative positions of the distribution of influence scores of the noise image tasks and the normal tasks to change depending on the extent of generalization.

### B.3.2 PROTOTYPICAL NETWORK

The experimental setup for Prototypical Network is parallel with the one for MAML. We trained four-layer CNN backbones (for 113,088 parameters for MiniImagenet and 111.936 parameters for Omniglot) for the 5-way-5-shot problem. Table 7, 8, 9, and 10 show the result of the experiment.

Table 7: Experiments with a CNN as a Prototypical network trained with MiniImagenet dataset combined with 128 noise image tasks. 128 test tasks were taken from the test taskset of pure MiniImagenet. See the caption of Table 4 for other notations.

| | training parameter | | | accuracy | | # test with proper score order | |
|---|---|---|---|---|---|---|---|
| # tasks | # normal tasks | # noise tasks | iteration | test | train | mean | median |
| 136 | 8 | 128 | 10000 | 0.357 | 1.000 | 91(+4.77$\sigma$) | 86(+3.89$\sigma$) |
| 144 | 16 | 128 | 10000 | 0.363 | 1.000 | 85(+3.71$\sigma$) | 79(+2.65$\sigma$) |
| 160 | 32 | 128 | 10000 | 0.333 | 1.000 | 66(+0.35$\sigma$) | 60(-0.71$\sigma$) |
| 192 | 64 | 128 | 10000 | 0.363 | 1.000 | 71(+1.24$\sigma$) | 70(+1.06$\sigma$) |
| 256 | 128 | 128 | 10000 | 0.393 | 0.999 | 24(-7.07$\sigma$) | 30(-6.01$\sigma$) |
| 512 | 384 | 128 | 10000 | 0.447 | 0.751 | 115(+9.02$\sigma$) | 110(+8.13$\sigma$) |
| 1024 | 896 | 128 | 10000 | 0.507 | 0.621 | 94(+5.30$\sigma$) | 92(+4.95$\sigma$) |
| 2024 | 1896 | 128 | 10000 | 0.535 | 0.599 | 88(+4.24$\sigma$) | 77(+2.30$\sigma$) |

Table 8: Experiment with a Prototypical network trained with MiniImagenet dataset and noise image tasks with a fixed mixing rate. 128 test tasks were taken from the test taskset of pure MiniImagenet. See the caption of Table 4 for other notations.

| | training parameter | | | accuracy | | # test with proper score order | |
|---|---|---|---|---|---|---|---|
| # tasks | # normal tasks | # noise tasks | iteration | test | train | mean | median |
| 128 | 112 | 16 | 10000 | 0.417 | 1.000 | 33(-5.48$\sigma$) | 38(-4.60$\sigma$) |
| 256 | 224 | 32 | 10000 | 0.434 | 0.916 | 122(+10.25$\sigma$) | 98(+6.01$\sigma$) |
| 512 | 448 | 64 | 10000 | 0.490 | 0.732 | 111(+8.31$\sigma$) | 101(+6.54$\sigma$) |
| 1024 | 896 | 128 | 10000 | 0.507 | 0.621 | 94(+5.30$\sigma$) | 92(+4.95$\sigma$) |

Table 9: Experiments with a CNN as a Prototypical network trained with Omniglot dataset combined with 128 noise image tasks. 128 test tasks were taken from the test taskset of pure Omniglot. See the caption of Table 4 for other notations.

| | training parameter | | | accuracy | | # test with proper score order | |
|---|---|---|---|---|---|---|---|
| # tasks | # normal tasks | # noise tasks | iteration | test | train | mean | median |
| 136 | 8 | 128 | 10000 | 0.679 | 1.000 | 81(+3.01$\sigma$) | 70(+1.06$\sigma$) |
| 144 | 16 | 128 | 10000 | 0.699 | 1.000 | 79(+2.65$\sigma$) | 83(+3.36$\sigma$) |
| 160 | 32 | 128 | 10000 | 0.719 | 1.000 | 64(+0.00$\sigma$) | 68(+0.71$\sigma$) |
| 192 | 64 | 128 | 10000 | 0.654 | 1.000 | 84(+3.54$\sigma$) | 75(+1.94$\sigma$) |
| 256 | 128 | 128 | 10000 | 0.697 | 1.000 | 86(+3.89$\sigma$) | 85(+3.71$\sigma$) |
| 512 | 384 | 128 | 10000 | 0.750 | 0.998 | 76(+2.12$\sigma$) | 70(+1.06$\sigma$) |
| 1024 | 896 | 128 | 10000 | 0.754 | 0.971 | 77(+2.30$\sigma$) | 72(+1.41$\sigma$) |
| 2024 | 1896 | 128 | 10000 | 0.841 | 0.960 | 59(-0.88$\sigma$) | 55(-1.59$\sigma$) |

Table 10: Experiments with a CNN as a Prototypical network trained with Omniglot dataset combined with noise image tasks at the mixing rate 7:1. 128 test tasks were taken from the test taskset of pure Omniglot. See the caption of Table 4 for other notations.

| | training parameter | | | accuracy | | # test with proper score order | |
|---|---|---|---|---|---|---|---|
| # tasks | # normal tasks | # noise tasks | iteration | test | train | mean | median |
| 128 | 112 | 16 | 10000 | 0.727 | 1.000 | 70(+1.06$\sigma$) | 68(+0.71$\sigma$) |
| 256 | 224 | 32 | 10000 | 0.717 | 1.000 | 68(+0.71$\sigma$) | 68(+0.71$\sigma$) |
| 512 | 448 | 64 | 10000 | 0.740 | 1.000 | 68(+0.71$\sigma$) | 67(+0.53$\sigma$) |
| 1024 | 896 | 128 | 10000 | 0.754 | 0.971 | 77(+2.30$\sigma$) | 72(+1.41$\sigma$) |

Table 11: Experiments with a CNN trained with MAML and Omniglot dataset combined with 128 noise image tasks. 128 test tasks were taken from the combined training taskset. See the caption of Table 4 for other notations.

| | training parameter | | | accuracy | self-rank | # test with proper score order | |
|---|---|---|---|---|---|---|---|
| # tasks | # normal | # noise | iteration | train | | mean | median |
| 136 | 8 | 128 | 17000 | 1.00 | 0.00±0.00 | 10(-9.55$\sigma$) | 59(-0.88$\sigma$) |
| 144 | 16 | 128 | 9000 | 1.00 | 0.00±0.00 | 18(-8.13$\sigma$) | 60(-0.71$\sigma$) |
| 160 | 32 | 128 | 10000 | 1.00 | 0.00±0.00 | 27(-6.54$\sigma$) | 69(+0.88$\sigma$) |
| 192 | 64 | 128 | 18000 | 1.00 | 0.00±0.00 | 44(-3.54$\sigma$) | 60(-0.71$\sigma$) |
| 256 | 128 | 128 | 12000 | 1.00 | 0.00±0.00 | 60(-0.71$\sigma$) | 61(-0.53$\sigma$) |
| 512 | 384 | 128 | 20000 | 1.00 | 0.00±0.00 | 68(+0.71$\sigma$) | 56(-1.41$\sigma$) |
| 1024 | 896 | 128 | 18000 | 1.00 | 0.01±0.09 | 54(-1.77$\sigma$) | 46(-3.18$\sigma$) |
| 2048 | 1920 | 128 | 15000 | 1.00 | 10.73±72.39 | 66(+0.35$\sigma$) | 68(+0.71$\sigma$) |

Table 12: Experiments with a CNN trained with MAML and Omniglot dataset combined with noise image tasks at the mixing rate 7:1. 128 test tasks were taken from the combined training taskset. See the caption of Table 4 for other notations.

| | training parameter | | | accuracy | self-rank | # test with proper score order | |
|---|---|---|---|---|---|---|---|
| # tasks | # normal | # noise | iteration | train | | mean | median |
| 128 | 112 | 16 | 20000 | 1.00 | 0.00±0.00 | 106(+7.42$\sigma$) | 59(-0.88$\sigma$) |
| 256 | 224 | 32 | 20000 | 1.00 | 0.02±0.20 | 85(+3.71$\sigma$) | 52(-2.12$\sigma$) |
| 512 | 448 | 64 | 20000 | 1.00 | 0.00±0.00 | 77(+2.30$\sigma$) | 55(-1.59$\sigma$) |
| 1024 | 896 | 128 | 18000 | 1.00 | 0.01±0.09 | 54(-1.77$\sigma$) | 46(-3.18$\sigma$) |

The results of the test accuracies with Omniglot (Table 9 and 10) show that the decrease of the generalization ability due to the number of training tasks is not as severe as with MAML. This is reflected in the relative positions of the training task distributions, as measured by the means and medians of influence scores, which remain relatively stable. In contrast, the results with MiniImagenet (Table 7 and 8) both show test accuracy decreases with the number of training tasks and the shift in the relative positions occues in the regions with the number of normal tasks being around 120. Note that in the case of the fixed number of noise tasks (Table 7), further decreasing of the tasks leads to the change of the relative position again. This can be considered due to the shortage of statistics. In those situations, a small number of normal training tasks with high scores can affect the mean or median values significantly, and those statistical values are less adequate for characterizing the positions of the task distributions.

### B.4 TEST WITH TRAINING TASKS

To check if the CNNs used in the above experiments have Property 1 in Section 5, we tested them with tasks taken from the training tasks. The results are shown in Table 11, 12, 13, 14, 15, and 16.

The columns of self-ranks in the tables indicate that the influence scores with the approximation method also have the ability to distinguish the tasks identical to the test tasks both for MAML and

Table 13: Experiments with a CNN as a Prototypical network trained with MiniImagenet dataset combined with 128 noise image tasks. 128 test tasks were taken from the combined training taskset. See the caption of Table 4 for other notations.

| | training parameter | | | accuracy | self-rank | # test with proper score order | |
|---|---|---|---|---|---|---|---|
| # tasks | # normal | # noise | iteration | train | | mean | median |
| 136 | 8 | 128 | 10000 | 1.00 | 0.00±0.00 | 8(-9.90$\sigma$) | 25(-6.89$\sigma$) |
| 144 | 16 | 128 | 10000 | 1.00 | 0.00±0.00 | 16(-8.49$\sigma$) | 28(-6.36$\sigma$) |
| 160 | 32 | 128 | 10000 | 1.00 | 0.00±0.00 | 29(-6.19$\sigma$) | 28(-6.36$\sigma$) |
| 192 | 64 | 128 | 10000 | 1.00 | 0.00±0.00 | 42(-3.89$\sigma$) | 22(-7.42$\sigma$) |
| 256 | 128 | 128 | 10000 | 1.00 | 0.00±0.00 | 56(-1.41$\sigma$) | 15(-8.66$\sigma$) |
| 512 | 384 | 128 | 10000 | 0.75 | 0.00±0.00 | 69(+0.88$\sigma$) | 55(-1.59$\sigma$) |
| 1024 | 896 | 128 | 10000 | 0.62 | 0.00±0.00 | 87(+4.07$\sigma$) | 72(+1.41$\sigma$) |
| 2024 | 1896 | 128 | 10000 | 0.60 | 0.00±0.00 | 84(+3.54$\sigma$) | 84(+3.54$\sigma$) |

Table 14: Experiments with a CNN as a Prototypical network trained with MiniImagenet dataset combined with noise image tasks at the mixing rate 7:1. 128 test tasks were taken from the combined training taskset. See the caption of Table 4 for other notations.

| # tasks | training parameter # normal | # noise | iteration | accuracy train | self-rank | # test with proper score order mean | median |
|---|---|---|---|---|---|---|---|
| 128 | 112 | 16 | 10000 | 1.00 | 0.00±0.00 | 103(+6.89$\sigma$) | 25(-6.89$\sigma$) |
| 256 | 224 | 32 | 10000 | 0.92 | 0.00±0.00 | 60(-0.71$\sigma$) | 22(-7.42$\sigma$) |
| 512 | 448 | 64 | 10000 | 0.73 | 0.00±0.00 | 80(+2.83$\sigma$) | 68(+0.71$\sigma$) |
| 1024 | 896 | 128 | 10000 | 0.62 | 0.00±0.00 | 87(+4.07$\sigma$) | 72(+1.41$\sigma$) |

Table 15: Experiments with a CNN as a Prototypical network trained with Omniglot dataset combined with 128 noise image tasks. 128 test tasks were taken from the combined training taskset. See the caption of Table 4 for other notations.

| # tasks | training parameter # normal | # noise | iteration | accuracy train | self-rank | # test with proper score order mean | median |
|---|---|---|---|---|---|---|---|
| 136 | 8 | 128 | 10000 | 1.00 | 0.00±0.00 | 12(-9.19$\sigma$) | 43(-3.71$\sigma$) |
| 144 | 16 | 128 | 10000 | 1.00 | 0.00±0.00 | 14(-8.84$\sigma$) | 42(-3.89$\sigma$) |
| 160 | 32 | 128 | 10000 | 1.00 | 0.04±0.26 | 27(-6.54$\sigma$) | 35(-5.13$\sigma$) |
| 192 | 64 | 128 | 10000 | 1.00 | 1.09±6.42 | 45(-3.36$\sigma$) | 34(-5.30$\sigma$) |
| 256 | 128 | 128 | 10000 | 1.00 | 3.67±16.55 | 57(-1.24$\sigma$) | 81(+3.01$\sigma$) |
| 512 | 384 | 128 | 10000 | 1.00 | 30.11±56.72 | 69(+0.88$\sigma$) | 83(+3.36$\sigma$) |
| 1024 | 896 | 128 | 10000 | 0.97 | 83.41±122.60 | 66(+0.35$\sigma$) | 64(+0.00$\sigma$) |
| 2024 | 1896 | 128 | 10000 | 0.96 | 120.10±241.10 | 54(-1.77$\sigma$) | 63(-0.18$\sigma$) |

Table 16: Experiments with a CNN as a Prototypical network trained with Omniglot dataset combined with noise image tasks at the mixing rate 7:1. 128 test tasks were taken from the combined training taskset. See the caption of Table 4 for other notations.

| # tasks | training parameter # normal | # noise | iteration | accuracy train | self-rank | # test with proper score order mean | median |
|---|---|---|---|---|---|---|---|
| 28 | 112 | 16 | 10000 | 1.00 | 1.70±9.24 | 109(+7.95$\sigma$) | 73(+1.59$\sigma$) |
| 256 | 224 | 32 | 10000 | 1.00 | 7.12±23.36 | 86(+3.89$\sigma$) | 48(-2.83$\sigma$) |
| 512 | 448 | 64 | 10000 | 1.00 | 36.98±69.89 | 55(-1.59$\sigma$) | 54(-1.77$\sigma$) |
| 1024 | 896 | 128 | 10000 | 0.97 | 83.41±122.60 | 66(+0.35$\sigma$) | 64(+0.00$\sigma$) |

Prototypical Network. However, we should note that self-ranks deviate slightly from 0 in some experiments with Prototypical Network and Omniglot dataset. A possible interpretation is that Omniglot dataset has pairs of closely similar tasks that the influence score fail to distinguish. This may contribute to why tasks in the Omniglot dataset are easier to learn than those in MiniImagenet.

We should also mention the counts of the tests with the proper order of the normal task distributions and the noise task distributions. In the cases of the combined tasksets with the fixed mixing ratio (Table 12, 14, and 16 ). We observe that the mean scores of normal tasks tend to increase relative to those of noise tasks when the number of training tasks is reduced, although this tendency is less pronounced in the median scores. The tendency of the mean values is considered due to their susceptibility to a small number of high scores. The relatively weak tendency of the median values can be viewed as the result of their smaller susceptibility to those high scores.

In the cases of the combined tasksets with the fixed number of noise tasks (Table 11, 13, and 15 ), we observe the tendencies opposite to the above. In those cases, the mean scores of normal tasks tend to decrease when the number of training tasks is reduced. Similarly, the median scores show less tendency. The reason is that the test tasks also contain the same portion of noise tasks. When the number of training tasks is reduces, most of the test tasks are noise tasks. For those noise test tasks, the mean scores of noise tasks become large, which means the mean scores of normal tasks become small.

### B.5 VALIDITY OF THE HESSIAN APPROXIMATION

From the above results, we see that influence scores calculated with GN matrix approximation have enough expressibility for finding training tasks identical to test tasks and some statistical values over different tests can also be used to distinguish the task distributions. As reference information for observing the extent to which those results are under the influence of the approximation, we trained a 2-layer fully connected network of the same structure as in Section B.2.1 with MAML and Omniglot and calculated the correlation coefficients of influence scores obtained with and without the approximation. As training settings, we took 5 cases of different numbers of training tasks, and for each of them, as conditions for inverting the Hessians, we took 5 cases of different numbers of the exact Hessian eigenvalues treated as non-zero, and 5 cases of different vector buffer sizes for the approximated Hessians. In Table 17, 18, 19,20, and 21, we list the mean $\pm$ std values resulting from calculating correlation coefficients between the influence scores of the training tasks and taking statistics over 128 tests for each setting.

From the tables, we observe that most of the diagonal elements have the largest values in the rows and in the columns where they reside. This is because the number of inverted eigenvalues in the exact method and the size of the vector buffer in the approximation method both determine how many directions are treated as non-flat, establishing a correspondence between the two.

We note that columns of larger inverted eigenvalues and rows of larger vector buffers exhibit weaker correlations. This is likely due to instability in inversion caused by small eigenvalues. Additionally, the network used here is much smaller than the CNN used in the previous subsections. We can expect that larger networks fit better to the training data and therefore, the approximated influence scores have more correlations with the exact ones. However, we cannot verify this experimentally due to the computational barriers and the memory cost.

Table 17: 64 training tasks, accuracy: 0.85

| GN Matrix | # inverted eigenvalues in exact Hessian | | | | |
|---|---|---|---|---|---|
| vector buffer | 64 | 128 | 256 | 512 | 1024 |
| 64 | 0.45±0.26 | 0.32±0.35 | 0.30±0.38 | 0.18±0.44 | 0.03±0.41 |
| 128 | 0.34±0.30 | 0.36±0.37 | 0.37±0.39 | 0.23±0.51 | 0.03±0.46 |
| 256 | 0.21±0.34 | 0.27±0.39 | 0.30±0.41 | 0.26±0.50 | 0.04±0.42 |
| 512 | 0.19±0.34 | 0.25±0.40 | 0.30±0.41 | 0.28±0.48 | 0.06±0.38 |
| 1024 | 0.19±0.34 | 0.25±0.39 | 0.30±0.39 | 0.27±0.46 | 0.07±0.36 |

Table 18: 128 training tasks, accuracy: 0.84

| GN Matrix | # inverted eigenvalues in exact Hessian | | | | |
|---|---|---|---|---|---|
| vector buffer | 64 | 128 | 256 | 512 | 1024 |
| 64 | 0.48±0.14 | 0.36±0.14 | 0.23±0.11 | 0.13±0.11 | 0.01±0.09 |
| 128 | 0.34±0.11 | 0.42±0.11 | 0.32±0.12 | 0.19±0.11 | 0.03±0.12 |
| 256 | 0.19±0.11 | 0.27±0.11 | 0.38±0.10 | 0.28±0.11 | 0.03±0.13 |
| 512 | 0.08±0.10 | 0.13±0.12 | 0.22±0.10 | 0.29±0.10 | 0.07±0.16 |
| 1024 | 0.06±0.13 | 0.08±0.12 | 0.13±0.11 | 0.20±0.11 | 0.07±0.15 |

Table 19: 256 training tasks, accuracy: 0.87

| GN Matrix | # inverted eigenvalues in exact Hessian | | | | |
|---|---|---|---|---|---|
| vector buffer | 64 | 128 | 256 | 512 | 1024 |
| 64 | 0.44±0.11 | 0.35±0.10 | 0.24±0.10 | 0.15±0.10 | 0.02±0.11 |
| 128 | 0.34±0.10 | 0.42±0.09 | 0.32±0.10 | 0.20±0.09 | 0.03±0.10 |
| 256 | 0.23±0.09 | 0.31±0.08 | 0.38±0.08 | 0.28±0.09 | 0.04±0.10 |
| 512 | 0.14±0.09 | 0.18±0.09 | 0.25±0.08 | 0.31±0.08 | 0.06±0.09 |
| 1024 | 0.09±0.08 | 0.11±0.09 | 0.15±0.08 | 0.21±0.09 | 0.10±0.09 |

Table 20: 512 training tasks, accuracy: 0.88

| GN Matrix | # inverted eigenvalues in exact Hessian | | | | |
|---|---|---|---|---|---|
| vector buffer | 64 | 128 | 256 | 512 | 1024 |
| 64 | 0.39±0.10 | 0.32±0.10 | 0.23±0.09 | 0.16±0.07 | 0.02±0.07 |
| 128 | 0.31±0.07 | 0.38±0.07 | 0.30±0.08 | 0.21±0.07 | 0.03±0.07 |
| 256 | 0.22±0.07 | 0.29±0.06 | 0.36±0.07 | 0.29±0.06 | 0.04±0.08 |
| 512 | 0.15±0.07 | 0.20±0.06 | 0.27±0.07 | 0.33±0.05 | 0.06±0.07 |
| 1024 | 0.10±0.06 | 0.13±0.06 | 0.19±0.07 | 0.25±0.06 | 0.08±0.06 |

Table 21: 1024 training tasks, accuracy: 0.89

| GN Matrix | # inverted eigenvalues in exact Hessian | | | | |
|---|---|---|---|---|---|
| vector buffer | 64 | 128 | 256 | 512 | 1024 |
| 64 | 0.36±0.10 | 0.29±0.09 | 0.22±0.08 | 0.15±0.07 | 0.02±0.05 |
| 128 | 0.31±0.07 | 0.36±0.07 | 0.30±0.08 | 0.22±0.06 | 0.04±0.05 |
| 256 | 0.23±0.06 | 0.31±0.06 | 0.36±0.06 | 0.29±0.05 | 0.06±0.05 |
| 512 | 0.16±0.05 | 0.22±0.05 | 0.29±0.05 | 0.33±0.05 | 0.08±0.05 |
| 1024 | 0.12±0.05 | 0.16±0.05 | 0.21±0.05 | 0.27±0.04 | 0.11±0.05 |

