# OpenReview forum: "TLXML: Task-Level Explanation of Meta-Learning via Influence Functions"
_ICLR.cc/2025/Conference — Submitted to ICLR 2025_

### Official Review · Reviewer_btUn · 2024-10-23

**Soundness:** 3
**Presentation:** 2
**Contribution:** 2
**Rating:** 5
**Confidence:** 5

**Summary:**

This paper propose to apply the Influence Function methods to meta-learning to analyze the influence of each training task to meta-testing tasks.

**Strengths:**

The proposed method is technically sound. It is reasonable that transferring the Influence Function from supervised-learning to meta-learning can help analyzing the influence of meta-training tasks on meta-testing tasks.

**Weaknesses:**

The meaning of analyzing task-level influence from meta-training to meta-testing is not well-understood. It is blurred why TLXML can bring advantage to XmetaAI. The paper shows its motivation by arguing that existing works of XmetaAI are some what "local", but why people need more "global" explanation is neither introduced persuasively nor empirically shown specifically.

Technically, TLXML seems like an implemental practice of the 'supervised-learning'-'meta-learning' mapping proposed in [1], where he objective is the Influence Function.

There lacks empirical results to straightforwardly measure the correlation of the value of eqn(7) and the true influence, to essentially support the correctness of the TLXML.

[1] Chao, Wei-Lun, et al. "Revisiting meta-learning as supervised learning." arXiv preprint arXiv:2002.00573 (2020).

**Questions:**

1. Could the authors illustrate how TLXML helps XmetaAI more specifically?
2. Could the authors provide empirical results about the correlation of the value of eqn(7) and the testing performance removing corresponding meta-training task?

---

> ### Author Response · Authors · 2024-11-20
> **Comments on the weaknesses and answers to the questions**
>
> # Comments on the weaknesses
> ## Weakness 1 The meaning of analyzing task-level influence from meta-training to meta-testing
> A possible clarification is that an explanation method based on past experiences can utilize the structures of the training dataset. For example, in the case of Figure 3b, the second half of the explanation, "distinguish cars and ambulances based on general features of vehicle shape" is possible owing to the nature that training data can have attributes with various levels of abstraction, which can be added by annotators, or taken from descriptions of the data acquisition process. In contrast, extracting those attributes from each inference data point arising in the user environment is hard.
> We will add this as a comment around the 3rd paragraph of the introduction in the next version.
> Please also note the following. TLXML is an explanation method based on training tasks, and hence, it is something other than local methods because it can relate each inference task to training tasks, but we avoid saying that it is global because that explanation is still about each inference and not about the entire logic of the model.
> ## Weakness 2 Novelty of TLXML
> Thank you for directing us to the paper by [1] Chao, Wei-Lun, et al.
> Our work has a structure similar to [1] in the sense that it starts with viewing supervised learning and meta-learning with the same footing to extend a method of the former to the latter.
> However, it is just a starting point for defining the influence function (7).
>  This work provides more than that. As far as we know, it is the first work that motivates using influence functions to explain meta-learning. Moreover, it presents the definition that enables task-level explanation, reveals that it involves computational difficulties that do not exist in supervised learning, and proposes a method to mitigate them.
> ## Weakness 3 Empirical result on the correlation of the value of eqn(7) and the true influence
> The influence function (7), defined as the differentiation with respect to task perturbation parameters, has by itself a meaning as a sensitivity to the increase of the portion of each task in the task distribution, and we can examine its effectiveness without mentioning other definitions. Therefore, we consider that investigating its correlation to the one defined by the performance change caused by dropping each training task is not a subject of top priority. However, from a theoretical viewpoint, it is also interesting for us to see how well (7) approximates the ones with the other definition. We will conduct experiments about this if time allows.
> # Answers to the questions
> ## Question 1
> See the comment on Weakness 1.
> ## Question 2
> See the comment on Weakness 3.

---

> > ### Comment · Reviewer_btUn · 2024-11-28
> > **Thanks for your response**
> >
> > Thanks for the authors' response. I have raised my score.

---

### Official Review · Reviewer_djck · 2024-10-28

**Soundness:** 3
**Presentation:** 3
**Contribution:** 3
**Rating:** 5
**Confidence:** 3

**Summary:**

The paper proposes a method called Task-Level explanation of Meta-Learning (TLXML) to explain the behavior of meta-learning models through influence functions. Meta-learning allows models to adapt to new tasks, but it can also lead to inappropriate model updates in user environments, which raises safety concerns - for this reason it would be useful if one could argue how much each meta-task influences the prediction. TLXML uses influence functions to measure this impact of previously learned tasks on the model's behavior in a new task. The method provides concise, task-based explanations in order to facilitate better interpretability of meta-learning processes. The authors also propose an approximation method for the Hessian matrix of the training loss using the Gauss-Newton matrix, which reduces the computational cost from O(pq2) to O(pq), where p is the number of weights and q the number of meta-parameters. The proposed method is analyzed experiments on image classification tasks with MAML and Prototypical Network.

**Strengths:**

The paper is very well written and easy to follow. The meta-learning motivation, as well as literature analysis is (up to my knowledge) well done - I could not find any works that propose to use influence functions for meta learning. Furthermore, the problem is well motivated and it is clear why would a framework like TLXML benefit the meta-learning community. The problems that arise (e.g., computational complexity) are well formulated, through rigorous mathematical expressions. The authors propose a method to tackle this, using the Gauss-Newton matrix and give theoretical reasoning (as well as refer to related works that use similar approach). The authors evaluate their method with both Prototypical network as well as MAML and thoroughly explain how they obtain their final objective, going form task-level influence functions to group-based influence functions. The authors are also open about the weaknesses of some of the assumptions that they have to make (e.g., when having to calculate or invert the Hessian). They provide an analysis of pruning the Hessian and show that doing some pruning improves the results significantly and that TLXML is able to pick up the identical task in all of the attempts with pruning. They also show that the self ranks, across 128 tests, follow a distribution that seems to have exponentially fast decay, highlighting that TLXML might indeed be picking up the identical task most of the time.

**Weaknesses:**

I find the approach novel, well-motivated, and well theoretically situated. However, I find the experimental results unconvincing - I hope that the authors might be able to prove me wrong and improve them.

In the beginning of the paper, in Figure 1, the authors give a very nice key insight into TLXML, motivate it well, and leave the reader wanting to know how they solve this problem: calculating the influence on the model's behavior by providing which of the previously learned tasks are strongly/somewhat/not influential. Furthermore, the figure suggests the usage of TLXML with language models, which seems like the application within the LLM community might be fruitful and popular. However, coming to the actual results, this insight falls short. The authors put forward two properties:
Property 1: If the network memorizes a training task, the influence of the test task with similar characteristics should be higher than the others.
Property 2: The network encodes information about the training task distributions, and whether tasks sampled from distributions similar to the test task have a higher measured influence.

In section 5.1, the distinction of tasks is a very nice sanity check and shows that the method has the potential to work well. Furthermore, the pruning of the Hessian shows that the relevant tasks are consistently ranked first, which gives a very strong signal in the positive direction, showing the potential of the work. However, I do believe that 5.1 only gives a sanity check that TLXML passes - being able to figure out that the exact task at hand exists at the training dataset. It does validate property 1, but in reality, property 2 is much more important.

Then, in section 5.2, the authors attempt to validate Property 2, where they construct a set of training tasks sampled from different distributions and mix these with noise distributions. They do so to see whether the network encodes information about the training task distributions, and whether tasks sampled from distributions similar to the test task have a higher measured influence. This is a very valid point, but the experiment does not, in my opinion, show this. The experiments merely show that the network does not encode information from the tasks containing noise. I believe this might be exactly why the distributions of the scores for both regular and noise images overlap and do not show any clear distinction. To circumvent this, the authors propose to look at "proper order", but this is a much less stringent condition that shows that their framework indeed works and I believe a bit too weak condition - what proper order more shows is that, as mentioned before, the network does not pick up information from noise. I think the results were further weakened when the authors point out, in line 465 and 466, that class 1 of the test task is similar to 3 (out of five!) classes in the 1024th training task. The authors attribute this to the model trained on these classes to classify class 1 correctly without giving any proof for this. I would really like to see whether this is the case or whether this is actually showing that TLXML might not be the best choice for this dataset. If the authors can show that indeed training on classes 0, 1, and 2 of the 1024th task might indicate the model struggling to classify class 1 of the test task, then I believe this would be a very strong and positive result, but like this, it just remains unconvincing.


To summarize, regarding the analysis of property 1, I suggest that it be much shortened and the rest moved to the appendix, while outlining that the equations 4, 6, and 7 hold in the main part of the paper. The analysis of property 2 should be extended and indeed show the strange observation about the 1024th task and the test task.


Also, in Table 1, the accuracy on the test set is significantly smaller than training accuracy across all rotation angles, weight decays, and iterations. Although the authors might have wanted to test what happens when the model is overfitting, it would be beneficial to have the same table (and the written conclusions) for experiments where the CNN with MAML has not overfitted and see whether the conclusions still hold or not. In Appendix Tables 5 and 6, they show this for the Omniglot dataset, where non-overfitting can be observed only for a large number of tasks, but as they increase the number of tasks, the number of tests with proper score order does not increase, which gives the reader doubts about the solidity of the method. The conclusion is similar for the Prototypical network experiments in the appendix. Maybe the authors could try another dataset or at least give a bit more insight about this result.


To summarize, the paper lacks empirical evidence to support its key claim. I think the paper would be much stronger if the authors were able to provide some sort of semantic analysis of the tasks and show that the ones more semantically similar to the test task are valued higher in the TLXML framework, more akin to what was said in Figure 1. For example, the authors could check whether images of animals are valued more than images of cars when predicting animals, and vice versa. Another example could potentially be another dataset, where there is some pre-defined (or can be easily extracted) semantic meaning of the tasks, which can be compared to that of the test task. If the authors provide analysis such as this, as well as experimental results showcasing that classes 0 1 and 2 in 1024th task are indeed not informative about class 1 (or similar analysis to this, it does not have to be exactly for this experiment), I would be willing to increase my score and recommend acceptance.

**Questions:**

Here I give some minor details that I would like the authors to address.

The mathematics are clearly written but some of the notation is a bit confusing. For example, the authors introduce influence as $I^{param}$ and $I^{perf}$ but do not mention what these stand for. Could the authors briefly elaborate this before or just after introducing this notation. Also, could the same be done for $D^{trg}$, $D^{src}$? To be specific, I am talking about Section 3 (Preliminaries), the supervised meta-learning section.

In line 159, I do not see why does $A(T,w)$ not take the loss as input, but in line 165 it does? Could you explain your reasoning behind this - is this intended or is it an oversight maybe?

Could the authors please rephrase the paragraph in line 219, for now it seems a bit contradictory (maybe I misunderstood it)? At the beginning the authors mention that task-level explanations are insufficient alone, and that because of this they look at task grouping, however then just afterwards they mention task augmentation, an approach which is common when training models with limited number of tasks. If we have a limited number of tasks, why would we benefit then from task grouping? Why wouldn't in this case looking at task-level be better than looking at task-group level?

In Table 1, could the authors briefly please explain what conclusion can we draw from the column alpha/rank? It seems that there is no significant difference between any of the values (across all #eigenvalues). Could you provide a brief interpretation of these results or explain why this lack of significant difference is noteworthy?

In line 400 they mention third to seventh column and you have only 6 columns. Could the authors either correct the column count or clarify if there's a missing column that should be included?

In line 407 the authors mention 128 tasks with noise images with 996 training tasks, in total of 1024 tasks. The tasks don't seem to add up.  Could the authors verify these numbers and explain the discrepancy if it's not a typo? Thank you.

Further small typos:
Line 93 typo: explain -> explaining
Line 121 typo: based -> Based
Line 262 typo: metrc -> metric
Line 267 typo: cross-entropy should have space afterwards
Line 350 typo: validated -> validate
Line 365 please cite Bag of visual words and SIFT descriptor.

---

> ### Author Response · Authors · 2024-11-26
> **Comments on the weaknesses and answers to the questions**
>
> First of all, thank you for your careful reading and detailed comments.
>
> # Comments on the weaknesses
>
> Thank you for suggesting additional analyses. We also think both the semantic analysis of the results and analyses on non-overfitting regions will make the paper stronger. Since this work is in the phase of checking the concept supported by the theoretical argument, we put a priority on the analyses of non-overfitting regions, rather than investigating what is happening inside TLXML. Unfortunately, we have to say time limitations do not allow us to conduct them before the deadline for modification of the paper. If we get a quick result, we will present it in the discussion.
>
> # Answers to the questions
>
> > The mathematics are clearly written but some of the notation is a bit confusing.
>
> We will add some explanations in Section 3 in the next update of the paper.
>
>
> > In line 159, I do not see why does $A(T, w)$ not take the loss as input, but in line 165 it does?
>
> This is not a mistake. We employ the definition such that $\mathcal{T}$ is the pair of a dataset and a loss function.
>
> > Could the authors please rephrase the paragraph in line 219, for now it seems a bit contradictory (maybe I misunderstood it)?
>
> This paragraph is not written well. We correct them in the next update.
> We meant task-level explanations are "sometimes" insufficient. This happens when the tasks used in the training are so near to each other that human intuition can not distinguish them. For example, when task augmentation is employed in training, the influence of each of the deformed tasks is not of interest; rather, the influence of the group consisting of the tasks made from a single original task will be more useful. This group influence is used in some of the experiments in 5.2. We will add a comment at the end of the first paragraph of 5.2.
>
> > In Table 1, could the authors briefly please explain what conclusion can we draw from the column alpha/rank?
>
> Let us explain how a rank changes when $\alpha$ increases from 0 to 1 when \# eigenvalues are properly tuned to reduce the instability. The value of the rank starts with 0, and because it takes discrete values, it does not change from 0 until $\alpha$ reaches enough near 1. It monotonously increases as we expected, but this good property is not captured perfectly by the correlation coefficient between $\alpha$ and rank because the plot of them forms an L-shaped line with x direction flipped, not a straight line.
>
> > In line 400 they mention third to seventh column and you have only 6 columns. Could the authors either correct the column count or clarify if there's a missing column that should be included?
>
> We correct them in the next update. (seventh -> sixth)
>
> > In line 407 the authors mention 128 tasks with noise images with 996 training tasks, in total of 1024 tasks. The tasks don't seem to add up. Could the authors verify these numbers and explain the discrepancy if it's not a typo?
>
> We correct them in the next update. (996 -> 896)
>
> > Further small typos: Line 93 typo: explain -> explaining Line 121 typo: based -> Based Line 262 typo: metrc -> metric Line 267 typo: cross-entropy should have space afterwards Line 350 typo: validated -> validate Line 365 please cite Bag of visual words and SIFT descriptor.
>
> We correct them in the next update.

---

> > ### Comment · Reviewer_djck · 2024-11-27
> >
> > Thank you for acknowledging that some paragraphs could be written more clearly and the reply. I will maintain my score.

---

### Official Review · Reviewer_ehik · 2024-11-04

**Soundness:** 2
**Presentation:** 2
**Contribution:** 2
**Rating:** 3
**Confidence:** 4

**Summary:**

The paper presents TLXML, an approach for explaining meta-learning models by quantifying the influence of training tasks on adaptation and inference. TLXML leverages influence functions to trace the sensitivity of training tasks to model updates, extending prior work on influence functions to a meta-learning context. The method includes an approximation technique for the Hessian matrix using the Gauss-Newton matrix to reduce computational costs. Experimental validation is conducted on MAML and Prototypical Network.

**Strengths:**

- The paper extends influence functions to task-level explanations in meta-learning, which has not been widely explored.

- The authors propose approximating the Hessian matrix with the Gauss-Newton matrix, which makes sense and reduces computation cost.

- By tackling the under-explored area of explainability in meta-learning, the paper can contribute to interpretable meta-learning.

**Weaknesses:**

1.	Complex and Unclear Motivation: The motivation is presented in a dense, complex manner with terms like “local explanations” and “moment of inference” introduced without sufficient explanation. Phrases like “dire consequences” are vague and could be more specific. This makes it difficult for readers to understand the motivation and connect the proposed method to its practical benefits.

2.	Lack of Comparison with a Simple Task Embedding Baseline: The paper does not compare its proposed influence functions with a simpler baseline, such as calculating similarity between task embeddings. Using task embeddings to represent each task and measure similarity to test tasks would provide an obvious, computationally efficient baseline. Without this comparison, it’s unclear whether the additional complexity of influence functions offers significant advantages.

3.	Unclear Storage Efficiency Claims: The authors claim that retaining only the influence measures ( I_{\text{meta}} ) reduces storage requirements, making the approach suitable for devices with limited capacity. However, since  I_{\text{meta}}  requires comparisons with training data for each new test task, some form of training data still needs to be stored. This inconsistency makes the storage efficiency claim appear misleading.

4.	Ambiguity in Task Grouping Method and Lack of Realistic Justification: Task grouping is introduced as a way to reduce complexity and provide higher-level explanations, but the paper lacks details on how task groups are formed. There is no mention of whether grouping relates to out-of-distribution (OOD) generalization, which would give the approach more practical relevance. Without a clear grouping methodology or OOD perspective, task grouping risks seeming arbitrary and less relevant.

**Questions:**

- Could the authors simplify the motivation for the paper, particularly the rationale behind focusing on task-level explanations for meta-learning? Additionally, could they clarify specific risks they aim to address (e.g., misunderstandings in safety-critical applications) and explain the limitations of “local explanations” in practical terms?

- Why did the authors choose influence functions over a simpler baseline, such as task embedding similarity? Could they provide a direct comparison to demonstrate that influence functions offer advantages in terms of interpretability or effectiveness?

- The paper claims to reduce storage requirements by retaining only  I_{\text{meta}} . Could the authors clarify how this approach mitigates storage needs, given that comparisons with training data are still necessary? Are there specific representations or summaries of training tasks that enable this efficiency?

- Could the authors elaborate on the motivation behind using task groups rather than individual task influence? Additionally, how are task groups formed in practice? Are they based on clustering or distributional characteristics? Is there an intended connection between task grouping and out-of-distribution generalization?

- Could the authors review the paper for grammar and presentation consistency? Simplifying language, ensuring correct subject-verb agreement, and defining key terms could make the paper more accessible and improve readability.

---

> ### Author Response · Authors · 2024-11-24
> **Comments on the weaknesses and answers to the questions**
>
> # Comment on the weaknesses
> ## Weakness 1 Complex and Unclear Motivation
> Thank you for pointing out the unclearness in presenting the motivation. We will modify the 2nd and the 3rd paragraph in the next version.
> ## Weakness 2 Lack of Comparison with a Simple Task Embedding Baseline
> Thank you for your suggestion. We also think comparing TLXML to task embedding methods is naturally the next step. Since this first work of TLXM is in the stage of checking the concept raised by theoretical arguments, we may face not just empirical but also theoretical matters involved by comparing methods with different readiness for application, but, if we get a quick result before the deadline for modification, we will include it in the next version.
> ## Weakness 3 Unclear Storage Efficiency Claims
> We disagree. TLXML reduces storage requirements in the user environment. We can calculate $I_{\text{meta}}$ without test tasks and prepare it immediately after the meta-learning phase. Owing to this compression of training task information, the raw data of the training tasks are not required in calculating the influence on the inference (e.g. $I_{\text{perf}}$) in the user environment.
> ## Weakness 4 Ambiguity in Task Grouping Method and Lack of Realistic Justification
> There are no general principles for defining task groups. We can freely define them depending on the situation. For example, if the training taskset has a hierarchical structure, it may be useful to group tasks and define the influence functions at a higher level. If we do task augmentations in training, in most cases, the influence of each of the deformed tasks is not of interest; rather the influence score of a group consisting of tasks made from a single original task will be more useful.
>
> We are not completely sure why the reviewer mentions OOD generalization, but it gives us an interesting topic. We expect that when OOD generalization happens, that means the model has extracted a high-level structure from the training taskset, and the group influence scores defined at that level might serve as a sign of OOD generalization.
>
>
> # Answers to the questions
> > Could the authors simplify the motivation for the paper, particularly the rationale behind focusing on task-level explanations for meta-learning? Additionally, could they clarify specific risks they aim to address (e.g., misunderstandings in safety-critical applications) and explain the limitations of “local explanations” in practical terms?
>
> See the comment on Weakness 1
>
> > Why did the authors choose influence functions over a simpler baseline, such as task embedding similarity? Could they provide a direct comparison to demonstrate that influence functions offer advantages in terms of interpretability or effectiveness?
>
> See the comment on Weakness 2
>
> > The paper claims to reduce storage requirements by retaining only I_{\text{meta}} . Could the authors clarify how this approach mitigates storage needs, given that comparisons with training data are still necessary? Are there specific representations or summaries of training tasks that enable this efficiency?
>
> See the comment on Weakness 3
>
> > Could the authors elaborate on the motivation behind using task groups rather than individual task influence? Additionally, how are task groups formed in practice? Are they based on clustering or distributional characteristics? Is there an intended connection between task grouping and out-of-distribution generalization?
>
> See the comment on Weakness 4
>
> > Could the authors review the paper for grammar and presentation consistency? Simplifying language, ensuring correct subject-verb agreement, and defining key terms could make the paper more accessible and improve readability.
>
> Sure. we will do them in the next version.

---

### Official Review · Reviewer_NwdD · 2024-11-04

**Soundness:** 3
**Presentation:** 3
**Contribution:** 4
**Rating:** 5
**Confidence:** 5

**Summary:**

This paper proposes to apply influence functions to the meta-learning setting to derive explanations for inference tasks from the tasks seen during fast adaptation. This approach of combining meta-learning and influence functions to explain a model's decision making seems quite novel. As the authors rightfully pointed it out, existing explainable meta-learning methods mostly overlook the peculiarities of meta-learning as they treat the model parameters as given or deliver only local explanations that don't capture prior experiences. The novelty of the approach comes from the fact the authors were able to extend the influence functions framework to the meta-learning setting. The authors propose a Task-Level Explanations framework (or TLXML) that provides task-based explanations that align with users’ abstraction levels. To scale their approach to complex models, they introduce an approximation method to compute the inversion of the expensive Hessian matrix that is needed to measure task-level influence on the performance of the meta-learner. This approximation makes the inversion more tractable as it reduces the computational cost from $\mathcal{O}(pq^{2})$ down to $\mathcal{O}(pq)$ where $p$ and $q$ denote respectively the weights of the model and the meta-parameters.

**Strengths:**

The strength of this paper lies in the novel formulation of influence functions for meta-learning. As highlighted above, the authors were able to extend the influence functions framework to the meta-learning setting by building upon the well-known influence functions objectives. The Task-Level Explanations framework (or TLXML) that the authors propose provides task-based explanations that align with users’ abstraction levels. To scale their approach to complex models, the authors introduce a tractable method, based on the Gauss-Newton matrix approximation method, for computing the inversion of the Hessian matrix that is needed to get task-level influences on the meta-learner. This approximation makes the inversion more tractable, yielding a reduction in computational cost from $\mathcal{O}(pq^{2})$ down to $\mathcal{O}(pq)$ where $p$ and $q$ denote respectively the weights of the model and the meta-parameters.

**Weaknesses:**

Although the mathematical framing of the problem is sound, the evaluation seems relatively weak. For instance, from the evaluation carried out by the authors, it is quite difficult to pinpoint which specific class(es) of instances among the set of fast-adapted tasks were influential in the decision rendered by the model on any given test instance from the test set of meta tasks. It would be indeed very helpful if the authors could rank by influence the classes that influenced the model in the decision it renders for every specific test task instance. That way, we can unequivocally see if the explanations make sense or not. Right now, the user is left figuring what task instance aligns with the decision made by the model. For instance, in Figure 6, we cannot say for sure if the class \emph{dog} played a role in the decision rendered by the model for the class \emph{worm} in the test task.

**Questions:**

It would help if the evaluation was a little more thorough as I highlighted it in the above section.

---

> ### Author Response · Authors · 2024-11-25
> **A Comment on the weakness and the question**
>
> Thank you for your suggestion. In this work, we motivated and introduced using influence functions for task-level explanations of meta-learning and discussed theoretically how to handle the difficulties involved by the first definition (4). We consider that this first work of TLXM is in the phase of checking the concept, and in that sense, investigating the influences of each class is something optional because it requires introducing class-level influence functions and discussion of the usefulness of task-level and class-level influence functions at the same time. To analyze what is happening in TLXML,
> it may be more appropriate to look for another dataset having class labels at higher levels. If we can decide on the setup of this further analysis and get a quick result before the deadline for modification, we will include it in the next version.

---

### Meta-Review · Area_Chair_hwWL · 2024-12-22

**Metareview:**

(a) The paper proposes TLXML, a method to explain meta-learning models by using influence functions to measure the impact of training tasks on adaptation and inference. It claims to provide task-based explanations and reduces computational cost through Hessian approximation. Findings include showing the potential of the method in task distinction and some analysis of task distribution distinction.

(b) The strengths are the novel application of influence functions to meta-learning, the proposed Hessian approximation method, and addressing an under-explored area of explainability in meta-learning.

(c) The weaknesses are complex and unclear motivation, lack of comparison with a simple baseline, unclear storage efficiency claims, ambiguity in task grouping method, unconvincing experimental results (especially in validating key properties), and lack of empirical evidence to support the main claim. Missing in the submission could be more in-depth semantic analysis of tasks and better experimental setups to show the effectiveness of the method.

(d) The main reasons for rejection are the significant weaknesses in the paper. The unclear motivation and lack of proper comparison make it hard to assess the novelty and value of the proposed method. The unconvincing experimental results and lack of empirical support for the key claims mean the paper fails to demonstrate the effectiveness of TLXML.

**Additional Comments On Reviewer Discussion:**

Reviewers raised several points. One reviewer noted the difficulty in pinpointing influential classes in the evaluation, and the authors responded that investigating class-level influence is optional at this concept-checking stage. Another reviewer pointed out issues like complex motivation, lack of baseline comparison, unclear storage claims, and task grouping problems. The authors addressed these by promising to simplify motivation, consider baseline comparison if possible, clarify storage claims, and explain task grouping. However, despite the authors' responses, the overall weaknesses remained significant. The lack of improvement in experimental results and the inability to provide stronger empirical evidence weighed heavily in the final decision to reject. The authors' responses did not fully address the concerns raised, and the paper still lacked the necessary depth and evidence to support its claims.

---

### Decision · Program_Chairs · 2025-01-22

Reject